# Self-Supervised Visual Representation Learning with Semantic Grouping

**Xin Wen**[1]    **Bingchen Zhao**[2,3]    **Anlin Zheng**[1,4]    **Xiangyu Zhang**[4]    **Xiaojuan Qi**[1]

[1]University of Hong Kong    [2]University of Edinburgh    [3]LunarAI    [4]MEGVII Technology

{wenxin, xjqi}@eee.hku.hk    zhaobc.gm@gmail.com
{zhenganlin, zhangxiangyu}@megvii.com

## Abstract

In this paper, we tackle the problem of learning visual representations from un-labeled scene-centric data. Existing works have demonstrated the potential of utilizing the underlying complex structure within scene-centric data; still, they commonly rely on hand-crafted objectness priors or specialized pretext tasks to build a learning framework, which may harm generalizability. Instead, we propose contrastive learning from data-driven semantic slots, namely SlotCon, for joint semantic grouping and representation learning. The semantic grouping is performed by assigning pixels to a set of learnable prototypes, which can adapt to each sample by attentive pooling over the feature and form new slots. Based on the learned data-dependent slots, a contrastive objective is employed for representation learning, which enhances the discriminability of features, and conversely facilitates grouping semantically coherent pixels together. Compared with previous efforts, by simultaneously optimizing the two coupled objectives of semantic grouping and contrastive learning, our approach bypasses the disadvantages of hand-crafted priors and is able to learn object/group-level representations from scene-centric im-ages. Experiments show our approach effectively decomposes complex scenes into semantic groups for feature learning and significantly benefits downstream tasks, including object detection, instance segmentation, and semantic segmentation. Code is available at: https://github.com/CVMI-Lab/SlotCon.

## 1   Introduction

Existing self-supervised approaches have demonstrated that visual representations can be learned from unlabeled data by constructing pretexts such as transformation prediction [22], instance dis-crimination [70, 31], and masked image modeling [2, 30, 67], *etc*. Among them, approaches based on instance discrimination [28, 6, 7, 11], which treat each image as a single class and employ a contrastive learning objective for training, have attained remarkable success and are beneficial to boost performance on many downstream tasks.

However, this success is largely built upon the well-curated object-centric dataset ImageNet [15], which has a large gap with the real-world data for downstream applications, such as city scenes [13] or crowd scenes [46]. Directly applying the instance discrimination pretext to these real-world data by simply treating the scene as a whole overlooks its intrinsic structures (*e.g.*, multiple objects and complex layouts) and thus will limit the potential of pre-training with scene-centric data [61]. This leads to our focus: learning visual representations from unlabeled scene-centric data.

Recent efforts to address this problem can be coarsely categorized into two types of research. One stream extends the instance discrimination task to pixel level for dense representation learning [54,

36th Conference on Neural Information Processing Systems (NeurIPS 2022).

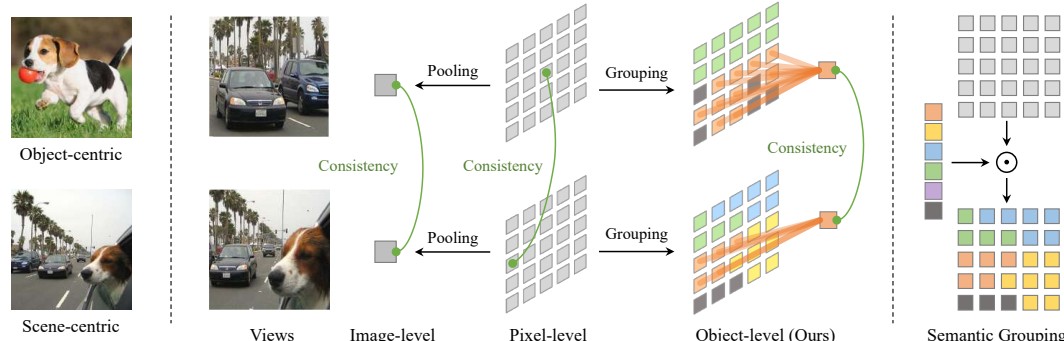

Figure 1: **Left:** Unlike object-centric images that highlight one iconic object, scene-centric images often contain multiple objects with complex layouts. This adds to the data diversity and increases the potential of image representations, yet it challenges previous learning paradigms that simply treat an image as a whole or individual pixels. **Middle:** Contrastive-learning objectives built upon different levels of image representations, in which object-level contrastive learning is viewed as the most suitable solution for scene-centric data. **Right:** We jointly learn a set of semantic prototypes in order to perform semantic grouping over the pixel-level representations and form object-centric slots.

66, 73], which shows strong performance in downstream dense prediction tasks. Yet, these methods still lack the ability to model object-level relationships presented in scene-centric data, which is crucial for learning representations. Although another stream of works attempts to perform object-level representation learning, most of them still heavily rely on domain-specific priors to discover objects, which include saliency estimators [62, 56], unsupervised object proposal algorithms [68, 72], hand-crafted segmentation algorithms [81, 34] or unsupervised clustering [35]. However, if the representation is supervised by hand-crafted objectness priors, it will be discouraged from learning objectness from the data itself and prone to mistakes from priors. Therefore, the capability and generalizability of the representation will be limited. In this work, we aim at a fully learnable and data-driven approach to enable learning representations from scene-centric data for enhanced effectiveness, transferability and generalizability.

We propose contrastive learning from data-driven semantic slots, namely SlotCon, for joint semantic grouping and representation learning. Semantic grouping is formulated as a feature-space pixel-level deep clustering problem where the cluster centers are initialized as a set of learnable semantic prototypes shared by the dataset, and grouping is achieved by assigning pixels to clusters. The cluster centers can then adapt to each sample by softly assigning pixels into cluster centers and aggregating their features via attentive pooling to form new ones, also called *slots*. Further, upon the learned slots from two random views of one image, a contrastive objective, which attempts to pull positive slots (*i.e.*, slots from the same prototype and sample) together and push away negative ones, is employed for representation learning. The optimized representations will enhance the discriminability of prototypes and slots, which conversely facilitates grouping semantically coherent pixels together. Compared with previous efforts, by simultaneously optimizing the two coupled objectives of semantic grouping and contrastive representation learning, our method bypasses the disadvantages of hand-crafted objectness priors[1] and is able to discover object/group-level representations from scene-centric images.

We extensively assess the representation learning ability of our model by conducting transfer learning evaluation on COCO [46] object detection, instance segmentation, and semantic segmentation on Cityscapes [13], PASCAL VOC [20], and ADE20K [83]. Our method shows strong results with both COCO pre-training and ImageNet-1K pre-training, bridging the gap between scene-centric and object-centric pre-training. As a byproduct, our method also achieves notable performance in unsupervised segmentation, showing strong ability in semantic concept discovery.

In summary, our main contributions in this paper are: 1) We show that the decomposition of natural scenes (semantic grouping) can be done in a learnable fashion and jointly optimized with the representations from scratch. 2) We demonstrate that semantic grouping can bring object-centric representation learning to large-scale real-world scenarios. 3) Combining semantic grouping and representation learning, we unleash the potential of scene-centric pre-training, largely close its gap with object-centric pre-training and achieve state-of-the-art results in various downstream tasks.

---

[1]Things can be different if the data is not reliable, see Section G in the appendix for details.

## 2 Related work

Our work is in the domain of self-supervised visual representation learning, where the goal is to learn visual representations without human annotations. We briefly review relevant works below.

**Image-level self-supervised learning** aims at learning visual representations by treating each image as one data sample. To this end, a series of pretext tasks are designed in which the labels are readily available without human annotations. Early explorations range from low-level pixel-wise reconstruction tasks that include denoising [64], inpainting [55], and cross-channel prediction [80] to higher-level instance discrimination [17], rotation prediction [22], context prediction [16], jigsaw puzzle [52], counting [53], and colorization [79]. Modern variants of instance discrimination [17] equipped with contrastive learning [60, 36] have shown strong potential in learning transferable visual representations [70, 8, 31, 58, 84, 82]. Other works differ in their learning objectives but still treat an image as a whole [28, 10, 78]. To further utilize the complex structure in natural images, some works exploit local crops [6, 71, 74, 61] while others either step to pixel- or object-level, detailed as follows.

**Pixel-level contrastive learning** extends the instance discrimination task from image-level feature vectors to feature maps [54, 47, 66, 73]. Their main differences lie in the way positive pixel-pairs are matched (spatial adjacency [54, 73], feature-space nearest-neighbor [66], sink-horn matching [47]), and the image-level baseline they build upon (MoCo v2 [54, 66], BYOL [47, 73]). Their pixel-level objective naturally helps learn dense representations that are favorable for dense prediction downstream tasks but lacks the grasp of holistic semantics and commonly require an auxiliary image-level loss to attain stronger performance [66, 73].

**Object-level contrastive learning** first discovers the objects in images and applies the contrastive objective over them, achieving a good balance in fine-grained structure and holistic semantics, yielding strong empirical gains with both object-centric [34, 35] and scene-centric data [72]. The key issue lies in finding objects in an image without supervision. Current works, however, still heavily rely on heuristic strategies that include saliency estimators [62, 56], selective-search [68, 72], hand-crafted segmentation algorithms [81, 34], or k-means clustering [35]. In contrast, our semantic grouping method is fully learnable and end to end, ensuring transferability and simplicity.

**Unsupervised semantic segmentation** is an emerging task that targets addressing semantic segmentation with only unlabeled images. The first attempt of IIC [40] maximizes the mutual information of augmented image patches, and later works [38, 62] rely on saliency estimators as a prior to bootstrap semantic pixel representations. Recently, PiCIE [39] adopt pixel-level deep clustering [5] to cluster the pixels into semantic groups, which SegDiscover [37] further improves by adopting super-pixels. On the other hand, Leopart [85], STEGO [29], and FreeSOLO [65] exploits pre-trained networks' attention maps for objectness. Still, they commonly rely on a (self-supervised) pre-trained network for initialization, while our method is trained fully from scratch.

**Object-centric representation learning** is viewed as an essential component of data-efficient, robust and interpretable machine learning algorithms [27]. Towards unsupervised object-centric representation learning, a series of works have been proposed [26, 3, 19, 48, 25, 24, 23]. Directly extracting objects from images is challenging due to the lack of supervision and thus prior works have long been restricted to synthetic data [42, 51, 1]. Recent works try to step toward real-world videos; however, they either adopt motion [75, 41] or depth [18] as a cue for objectness to solve this problem. Instead, our method shows that we can first explicitly optimize for desirable properties of clusters (that can describe an object) over the dataset, then retrieve the objects from an image with the learned prototypes. More importantly, it first shows the possibility of learning object-centric representations from large-scale unlabelled scene-centric natural images.

## 3 Method

### 3.1 Semantic grouping with pixel-level deep clustering

Given a dataset $\mathcal{D}$ of unlabeled images, we aim at learning a set of prototypes $\mathcal{S}$ that classifies each pixel into a meaningful group, such that pixels within the same group are semantic-consistent (have

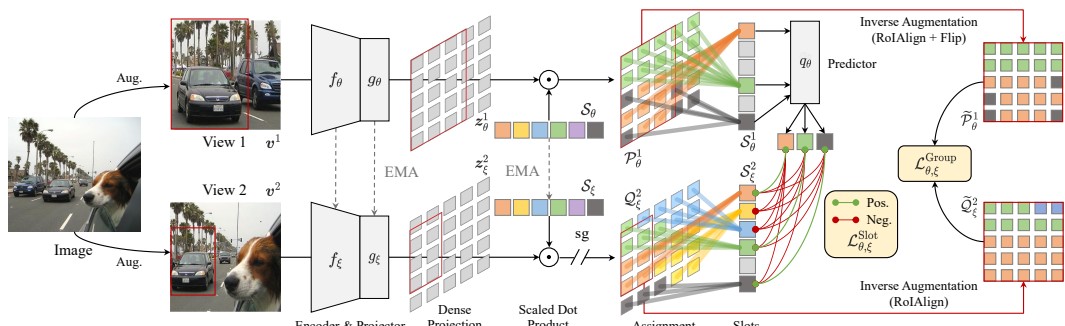

Figure 2: Overview of our proposed framework. Based on a shared pixel embedding function, the model learns to classify pixels into groups according to their feature similarity in a pixel-level deep clustering fashion (Sec. 3.1); the model produces group-level feature vectors (slots) through attentive pooling over the feature maps, and further performs group-level contrastive learning (Sec. 3.2). We omit the symmetrized loss computed by swapping the two views for simplicity. (*best viewed in color*)

similar feature representations), and pixels between different groups are semantic-incoherent. We find that this problem can be viewed as unsupervised semantic segmentation [22, 39], and solved with pixel-level deep clustering [5, 76, 6, 7].

Intuitively, a meaningful semantic grouping should be invariant to data augmentations. Thus, for different augmentations of the same image, we enforce the pixels that lie in the same location to have similar assignment scores w.r.t. the same set of cluster centers (prototypes). Besides consistent grouping, the groups should be different from each other to ensure that the learned representations are discriminative and avoid trivial solutions, *e.g.*, identical features. Together with standard techniques used in self-supervised learning (*e.g.*, non-linear projector and momentum teacher [28, 7], *etc.*), this leads to the following framework.

Specifically, as illustrated in Figure 2, our method contains two neural networks: the student network is parameterized by $\theta$ and include an encoder $f_\theta$, a projector $g_\theta$ and a set of $K$ learnable prototypes $\mathcal{S}_\theta = \left[ s_\theta^1, s_\theta^2, \ldots, s_\theta^K \right] \in \mathbb{R}^{K \times D}$; the teacher network holds the same architecture with the student, but uses a different set of weights $\xi$ that updates as an exponential moving average of $\theta$. Given an input image $x$, two random augmentations are applied to produce two augmented views $v^1$ and $v^2$. Each augmented view $v^l \in \{v^1, v^2\}$ is then encoded with an encoder $f$ into a hidden feature map $h^l \in \mathbb{R}^{H \times W \times D}$, and then transformed with a multilayer perceptron (MLP) $g$ to get $z^l = g\left(h^l\right) \in \mathbb{R}^{H \times W \times D}$. We then compute the assignment $\mathcal{P}_\theta^l$ of the projections $z_\theta^l$ with the corresponding prototypes $\mathcal{S}_\theta$, and enforce it to match the assignment $\mathcal{Q}_\xi^{l'}$ produced with another view $v^{l'}$ by the teacher network. More precisely, with $\ell_2$-normalized projections $\overline{z} = z/\|z\|$ and prototypes $\overline{s} = s/\|s\|$,

$$\mathcal{Q}_\xi^{l'} = \underset{K}{\text{softmax}} \left( \left( \overline{z}_\xi^{l'} \cdot \overline{\mathcal{S}}_\xi^\top - c \right) / \tau_t \right), \quad \mathcal{P}_\theta^l = \underset{K}{\text{softmax}} \left( \overline{z}_\theta^l \cdot \overline{\mathcal{S}}_\theta^\top / \tau_s \right) \in \mathbb{R}^{H \times W \times K}, \quad (1)$$

with $\tau_s, \tau_t > 0$ temperature parameters that control the sharpness of the output distribution of the two networks. The $c$ can be omitted and will be explained later. Note that the scale or layout of the two feature maps $z_\theta^l$ and $z_\xi^{l'}$ can be inconsistent due to geometric image augmentations, *e.g.*, random crop, scale or flip. So we perform the inverse augmentation process (including RoIAlign [32] and an optional flipping, details provided in Section A.1 in the appendix) on the predicted assignments to align their spatial locations: $\widetilde{\mathcal{Q}}_\xi^{l'} = \text{invaug}\left(\mathcal{Q}_\xi^{l'}\right), \widetilde{\mathcal{P}}_\theta^l = \text{invaug}\left(\mathcal{P}_\theta^l\right)$. The inverse augmentation is performed on the assignments $\mathcal{Q}$ rather than the projections $z$ to keep the context information out of the overlapping area for slot generation, which will be detailed in Section 3.2.

Based on the aligned assignments, we apply the cross-entropy loss $\mathcal{L}_{\theta,\xi}^{\text{CE}}\left(q_\xi, p_\theta\right) = -\sum_k q_\xi^k \log p_\theta^k$ to enforce the consistency in assignment score between spatial-aligned pixels from different views. The cross-entropy loss is averaged over all spatial locations to produce the grouping loss:

$$\mathcal{L}_{\theta,\xi}^{\text{Group}} = \frac{1}{H \times W} \sum_{i,j} \left[ \mathcal{L}_{\theta,\xi}^{\text{CE}}\left( \widetilde{\mathcal{Q}}_\xi^2[i,j], \widetilde{\mathcal{P}}_\theta^1[i,j] \right) + \mathcal{L}_{\theta,\xi}^{\text{CE}}\left( \widetilde{\mathcal{Q}}_\xi^1[i,j], \widetilde{\mathcal{P}}_\theta^2[i,j] \right) \right]. \quad (2)$$

Directly optimizing the above objective resembles an unsupervised variant of Mean Teacher [57], which collapses as shown in [28]. In order to avoid collapsing, we follow [7] to maintain a *mean logit* $c \in \mathbb{R}^K$ and reduce it when producing the teacher assignments $\mathcal{Q}_\xi$, as indicated in Eq. 1. The mean logit stores an exponential moving average of all the logits produced by the teacher network:

$$c \leftarrow \lambda_c c + (1 - \lambda_c) \frac{1}{B \times H \times W} \sum_{i,j,k} \overline{z}_\xi^{(i)}[j,k] \cdot \overline{\mathcal{S}}_\xi^\top , \tag{3}$$

where $B$ stands for the batch size. Intuitively, reducing the mean logit amplifies the difference in assignment between different pixels, preventing all pixels from being assigned to the same prototype. Besides that, the teacher temperature $\tau_t$ is smaller than the student temperature $\tau_s$ to produce a sharper target and avoid uniform assignments. Both operations help avoid collapse and force the network to learn a meaningful semantic grouping.

**Discussion with DINO.** The resulting solution for semantic grouping may seem like a naive extension of DINO [7]. However, this is not the whole picture. DINO is an *image-level* representation learning approach that adopts a large number of prototypes (*e.g.*, 65536), while our objective is built on *pixel-level* representations and is tailored to learn meaningful semantic groups. Intuitively, DINO captures scene semantics that can be more complex and thus requires more prototypes to represent all scene-level variations, while our method learns object-level semantics which can be composed to depict a complicated scene and thus only requires a small number of prototypes (*e.g.*, 256 for COCO). Besides, the representation learning objective of DINO is built on prototypes that are *shared by the whole dataset*, while our method instead adapts prototypes for each image and performs contrastive learning over groups (detailed in Section 3.2). In summary, we present a novel view for the decoupling of pixel-level online clustering (*i.e.*, semantic grouping) and object-level representation learning.

## 3.2   Group-level representation learning by contrasting slots

Inspired by Slot Attention [48], we then reuse the assignments computed by the semantic grouping module in Eq. 1 to perform attentive pooling over the dense projections $z$ to produce group-level feature vectors (rephrased as *slots*), as shown in Figure 2. Intuitively, as the softmax normalization applies to the slot dimension, the attention coefficients sum to one for each individual input feature vector. As a result, the soft assignments $\mathcal{A}$ of the dense projections $z$ w.r.t. the corresponding prototypes $\mathcal{S}$ can also be viewed as the attention coefficients that describe how the prototypes decompose the dense projections into non-overlapping groups. This inspires us to decompose the dense projections with attentive pooling following [48]. Specifically, for a dense projection $z_\theta^l$ produced from view $v^l$, we extract $K$ slots:

$$\mathcal{S}_\theta^l = \frac{1}{\sum_{i,j} \mathcal{A}_\theta^l[i,j]} \sum_{i,j} \mathcal{A}_\theta^l[i,j] \odot z_\theta^l[i,j] \in \mathbb{R}^{K \times D}, \mathcal{A}_\theta^l = \operatorname*{softmax}_K \left( \overline{z}_\theta^l \cdot \overline{\mathcal{S}}_\theta^\top / \tau_t \right) \in \mathbb{R}^{H \times W \times K} ,$$
$$\tag{4}$$

where $\odot$ denotes the Hadamard product, and a similar operation applies for the teacher network to produce $\mathcal{S}_\xi^{l'}$. Since the initial slots are shared by the whole dataset, the corresponding semantic may be missing in a specific view $v^l$, thus producing redundant slots. Therefore, we compute the following binary indicator $\mathbb{1}^l$ to mask out the slots that fail to occupy a dominating pixel:

$$\mathbb{1}_\theta^{k,l} = \exists_{i,j} \quad \text{such that} \quad \operatorname*{argmax}_K \left( \mathcal{A}_\theta^l \right)[i,j] = k , \tag{5}$$

and $\mathbb{1}_\xi^{l'}$ is computed similarly. Following the literature in object-level SSL [62, 34, 72, 35], we then apply a contrastive learning objective to discriminate the slot that holds the same semantic across views from distracting slots with the InfoNCE [60] loss:

$$\mathcal{L}_{\theta,\xi}^{\text{InfoNCE}} \left( \mathcal{S}_\theta^l, \mathcal{S}_\xi^{l'} \right) = \frac{1}{K} \sum_{k=1}^K - \log \frac{\mathbb{1}_\theta^{k,l} \mathbb{1}_\xi^{k,l'} \exp \left( \overline{q_\theta} \left( s_\theta^{k,l} \right) \cdot \overline{s}_\xi^{k,l'} / \tau_c \right)}{\sum_{k'} \mathbb{1}_\theta^{k,l} \mathbb{1}_\xi^{k',l'} \exp \left( \overline{q_\theta} \left( s_\theta^{k,l} \right) \cdot \overline{s}_\xi^{k',l'} / \tau_c \right)} . \tag{6}$$

This objective helps maximize the similarity between different views of the same slot, while minimizing the similarity between slots from another view with different semantics and *all* slots from other

images. Note that here an additional predictor $q_\theta$ with the same architecture as the projector $g_\theta$ is applied to the slots $\mathcal{S}_\theta$ as empirically it yields stronger performance [11, 68, 35]. And the resulting slot-level contrastive loss also follows a symmetric design like Eq. 2:

$$\mathcal{L}_{\theta,\xi}^{\text{Slot}} = \mathcal{L}_{\theta,\xi}^{\text{InfoNCE}}\left(\mathcal{S}_\theta^1, \mathcal{S}_\xi^2\right) + \mathcal{L}_{\theta,\xi}^{\text{InfoNCE}}\left(\mathcal{S}_\theta^2, \mathcal{S}_\xi^1\right) . \tag{7}$$

### 3.3 The overall optimization objective

We jointly optimize the semantic grouping objective (Eq. 2) and the group-level contrastive learning objective (Eq. 7), controlled with a balancing factor $\lambda_g$:

$$\mathcal{L}_{\theta,\xi}^{\text{Overall}} = \lambda_g \mathcal{L}_{\theta,\xi}^{\text{Group}} + (1 - \lambda_g)\mathcal{L}_{\theta,\xi}^{\text{Slot}} . \tag{8}$$

At each training step, the student network is optimized with gradients from the overall loss function: $\theta \leftarrow \text{optimizer}\left(\theta, \nabla_\theta \mathcal{L}_{\theta,\xi}^{\text{Overall}}, \eta\right)$, where $\eta$ denotes the learning rate; and the teacher network updates as an exponential moving average of the student network: $\xi \leftarrow \lambda_t \xi + (1 - \lambda_t)\theta$, with $\lambda_t$ denoting the momentum value. After training, only the teacher encoder $f_\xi$ is kept for downstream tasks.

## 4 Experiments

### 4.1 Implementation details

**Pre-training datasets.** We pre-train our models on COCO `train2017` [46] and ImageNet-1K [15], respectively. COCO `train2017` [46] contains ∼118K images of diverse scenes with objects of multiple scales, which is closer to real-world scenarios. In contrast, ImageNet-1K is a curated object-centric dataset containing

Table 1: Details of the datasets used for pre-training.

| Dateset | #Img. | #Obj./Img. | #Class |
|---|---|---|---|
| ImageNet-1K [15] | 1.28M | 1.7 | 1000 |
| COCO [46] | 118K | 7.3 | 80 |
| COCO+ [46] | 241K | N/A | N/A |

more than ∼1.28M images, which is better for evaluating a model's potential with large-scale data. Besides, we also explore the limit of scene-centric pre-training on COCO+, *i.e.*, COCO `train2017` set plus the `unlabeled2017` set. See details in Table 1.

**Data augmentation.** The image augmentation setting is the same as BYOL [28]: a $224 \times 224$-pixel random resized crop with a random horizontal flip, followed by a random color distortion, random grayscale conversion, random Gaussian blur, and solarization. The crop pairs without overlap are discarded during training.

**Network architecture.** We adopt ResNet-50 [33] as the default encoder for $f_\theta$ and $f_\xi$. The projector $g_\theta$, $g_\xi$ and predictor $q_\theta$ are MLPs whose architecture are identical to that in [7] with a hidden dimension of 4096 and an output dimension of 256.

**Optimization.** We adopt the LARS optimizer [77] to pre-train the model, with a batch size of 512 across eight NVIDIA 2080 Ti GPUs. Following [73], we utilize the cosine learning rate decay schedule [50] with a base learning rate of 1.0, linearly scaled with the batch size (LearningRate = $1.0 \times$ BatchSize$/256$), a weight decay of $10^{-5}$, and a warm-up period of 5 epochs. Following [72, 66, 73], the model is pre-trained for 800 epochs on COCO(+) and 100/200 epochs on ImageNet, respectively. Following the common practice of [72, 73], the momentum value $\lambda_t$ of the teacher model starts from 0.99 and is gradually increased to 1 following a cosine schedule. Synchronized batch normalization and automatic mixed precision are also enabled during training.

**Hyper-parameters.** The temperature values $\tau_s$ and $\tau_t$ in the student and teacher model are set to 0.1 and 0.07, respectively. Besides, the center momentum $\lambda_c$ is set to 0.9. The default number of prototypes $K$ is set to 256 for COCO(+) and 2048 for ImageNet, according to our empirical finding that setting the number of prototypes close to the number of human-annotated categories can help downstream performance. The temperature value $\tau_c$ for the contrastive loss is set to 0.2 following [11], and the default balancing ratio $\lambda_g$ is set to 0.5.

## 4.2 Evaluation protocols

Following the common practice of previous self-supervised works [66, 73, 72], we evaluate the representation ability of the pre-trained model by taking it as the backbone of downstream tasks. Specifically, we add a newly initialized task-specific head to the pre-trained model for different downstream tasks, *i.e.*, object detection and instance segmentation on COCO [46], and semantic segmentation on PASCAL VOC [20], Cityscapes [13], and ADE20K [83].

**Object detection and instance segmentation.** We train a Mask R-CNN [32] model with R50-FPN [45] backbone implemented in `Detectron2` [69]. We fine-tune all layers end-to-end on COCO `train2017` split with the standard $1\times$ schedule and report AP, $AP_{50}$, $AP_{75}$ on the `val2017` split. Following [66, 73, 72] we train with the standard $1\times$ schedule with SyncBN.

**Semantic segmentation.** The evaluation details of PASCAL VOC and Cityscapes strictly follow [31]. We take our network to initialize the backbone of a fully-convolutional network [49] and fine-tune all the layers end-to-end. For PASCAL VOC, we fine-tune the model on `train_aug2012` set for 30k iterations and report the mean intersection over union (mIoU) on the `val2012` set. For Cityscapes, we fine-tune on the `train_fine` set for 90k iterations and evaluate it on the `val_fine` set. For ADE20K, we follow the standard 80k iterations schedule of `MMSegmentation` [12].

**Unsupervised semantic segmentation.** We also evaluate the model's ability of discovering semantic groups in complex scenes, which is accomplished by performing unsupervised semantic segmentation on COCO-Stuff [4]. We follow the common practice in this field [40, 39, 37] to merge the labels into 27 categories (15 "stuff" categories and 12 "thing" categories), and evaluate with a subset created by [40]. We perform inference with resolution 320 and number of prototypes 27 following the standard practice. The predicted labels are matched with the ground truth through Hungarian matching [43], and evaluated on mIoU and pixel accuracy (pAcc).

## 4.3 Transfer learning results

**COCO pre-training.** In Table 2, we show the main results with COCO pre-training. There have been steady improvements in object-level pre-training with COCO, in which the top performance methods are DetCon [44] and ORL [72], which still rely on objectness priors like selective-search [59] or hand-crafted segmentation algorithms [21], yet they still fail to beat the pixel-level state-of-the-art

Table 2: **Main transfer results with COCO pre-training.** We report the results in COCO [46] object detection, COCO instance segmentation, and semantic segmentation in Cityscapes [13], PASCAL VOC [20] and ADE20K [83]. Compared with other image-, pixel-, and object-level self-supervised learning methods, our method shows consistent improvements over different tasks without leveraging multi-crop [6] and objectness priors. (†: re-impl. w/ official weights; ‡: full re-impl.)

| Method | Epochs | Multi crop | Obj. Prior | COCO detection | | | COCO segmentation | | | Semantic seg. (mIoU) | | |
|---|---|---|---|---|---|---|---|---|---|---|---|---|
| | | | | $AP^b$ | $AP_{50}^b$ | $AP_{75}^b$ | $AP^m$ | $AP_{50}^m$ | $AP_{75}^m$ | City. | VOC | ADE |
| random init. | - | ✗ | ✗ | 32.8 | 50.9 | 35.3 | 29.9 | 47.9 | 32.0 | 65.3 | 39.5 | 29.4 |
| *Image-level approaches* | | | | | | | | | | | | |
| MoCo v2‡ [9] | 800 | ✗ | ✗ | 38.5 | 58.1 | 42.1 | 34.8 | 55.3 | 37.3 | 73.8 | 69.2 | 36.2 |
| Revisit.† [61] | 800 | ✓ | ✗ | 40.1 | 60.2 | 43.6 | 36.3 | 57.3 | 38.9 | 75.3 | 70.6 | 37.0 |
| *Pixel-level approaches* | | | | | | | | | | | | |
| Self-EMD [47] | 800 | ✗ | ✗ | 39.3 | 60.1 | 42.8 | - | - | - | - | - | - |
| DenseCL† [66] | 800 | ✗ | ✗ | 39.6 | 59.3 | 43.3 | 35.7 | 56.5 | 38.4 | 75.8 | 71.6 | 37.1 |
| PixPro‡ [73] | 800 | ✗ | ✗ | 40.5 | 60.5 | 44.0 | 36.6 | 57.8 | 39.0 | 75.2 | **72.0** | 38.3 |
| *Object / Group-level approaches* | | | | | | | | | | | | |
| DetCon† [34] | 1000 | ✗ | ✓ | 39.8 | 59.5 | 43.5 | 35.9 | 56.4 | 38.7 | 76.1 | 70.2 | 38.1 |
| ORL† [72] | 800 | ✓ | ✓ | 40.3 | 60.2 | 44.4 | 36.3 | 57.3 | 38.9 | 75.6 | 70.9 | 36.7 |
| *Ours* (SlotCon) | 800 | ✗ | ✗ | **41.0** | **61.1** | **45.0** | **37.0** | **58.3** | **39.8** | **76.2** | 71.6 | **39.0** |

PixPro [73].[2] Our method alleviates such limitations and significantly improves over current object-level methods in all tasks, achieving consistent improvement over the previous approaches and even several methods that were pre-trained on the larger dataset ImageNet-1K (Table 4). It is also notable that our method can achieve a better performance on the largest and most challenging dataset for segmentation, ADE20K, adding to the significance of this work.

**COCO+ pre-training.**  In Table 3, we report the results with COCO+ pre-training. The COCO+ is COCO `train2017` plus `unlabeled2017` set, which roughly doubles the number of training images and greatly adds to the data diversity. Our method further sees a notable gain in all tasks with extended COCO+ data, and even shows comparable results with our best-performing model pre-trained on ImageNet-1K ($5\times$ large of COCO+), showing the great potential of scene-centric pre-training.

Table 3: **Pushing the limit of scene-centric pre-training.** Our method further sees a notable gain in all tasks with extended COCO+ data, showing the great potential of scene-centric pre-training.

| Method | Dataset | Epochs | COCO detection | | | COCO segmentation | | | Semantic seg. (mIoU) | | |
|---|---|---|---|---|---|---|---|---|---|---|---|
| | | | $AP^b$ | $AP^b_{50}$ | $AP^b_{75}$ | $AP^m$ | $AP^m_{50}$ | $AP^m_{75}$ | City. | VOC | ADE |
| SlotCon | COCO | 800 | 41.0 | 61.1 | 45.0 | 37.0 | 58.3 | 39.8 | 76.2 | 71.6 | 39.0 |
| SlotCon | ImageNet | 100 | 41.4 | 61.6 | 45.6 | 37.2 | 58.5 | 39.9 | 75.4 | 73.1 | 38.6 |
| SlotCon | ImageNet | 200 | **41.8** | **62.2** | 45.7 | **37.8** | 59.1 | **40.7** | 76.3 | **75.0** | 38.8 |
| ORL [72] | COCO+ | 800 | 40.6 | 60.8 | 44.5 | 36.7 | 57.9 | 39.3 | - | - | - |
| SlotCon | COCO+ | 800 | **41.8** | **62.2** | **45.8** | **37.8** | **59.4** | 40.6 | **76.5** | 73.9 | **39.2** |

**ImageNet-1K pre-training.**  In Table 4, we also benchmark our method with ImageNet-1K pre-training and show its compatibility with object-centric data. Without selective-search for object proposals and without pre-training and transferring the FPN head, our method still beats most of the current works and largely closes the gap with the detection-specialized method SoCo.

Table 4: **Main transfer results with ImageNet-1K pre-training.** Our method is also compatible with object-centric data and shows consistent improvements over different tasks without using FPN [45] and objectness priors. (†: re-impl. w/ official weights; ‡: full re-impl.)

| Method | Epochs | w/ FPN | Obj. Prior | COCO detection | | | COCO segmentation | | | Semantic seg. (mIoU) | | |
|---|---|---|---|---|---|---|---|---|---|---|---|---|
| | | | | $AP^b$ | $AP^b_{50}$ | $AP^b_{75}$ | $AP^m$ | $AP^m_{50}$ | $AP^m_{75}$ | City. | VOC | ADE |
| random init. | - | ✗ | ✗ | 32.8 | 50.9 | 35.3 | 29.9 | 47.9 | 32.0 | 65.3 | 39.5 | 29.4 |
| supervised | 100 | ✗ | ✗ | 39.7 | 59.5 | 43.3 | 35.9 | 56.6 | 38.6 | 74.6 | 74.4 | 37.9 |
| *Image-level approaches* | | | | | | | | | | | | |
| MoCo v2† [9] | 800 | ✗ | ✗ | 40.4 | 60.1 | 44.2 | 36.5 | 57.2 | 39.2 | 76.2 | 73.7 | 36.9 |
| DetCo† [71] | 200 | ✗ | ✗ | 40.1 | 61.0 | 43.9 | 36.4 | 58.0 | 38.9 | 76.0 | 72.6 | 37.8 |
| InsLoc† [74] | 200 | ✓ | ✗ | 40.9 | 60.9 | 44.7 | 36.8 | 57.8 | 39.4 | 75.4 | 72.9 | 37.3 |
| *Pixel-level approaches* | | | | | | | | | | | | |
| DenseCL† [66] | 200 | ✗ | ✗ | 40.3 | 59.9 | 44.3 | 36.4 | 57.0 | 39.2 | 76.2 | 72.8 | 38.1 |
| PixPro† [73] | 100 | ✗ | ✗ | 40.7 | 60.5 | 44.8 | 36.8 | 57.4 | 39.7 | **76.8** | 73.9 | 38.2 |
| *Object / Group-level approaches* | | | | | | | | | | | | |
| DetCon [34] | 200 | ✗ | ✓ | 40.6 | - | - | 36.4 | - | - | 75.5 | 72.6 | - |
| SoCo‡ [68] | 100 | ✓ | ✓ | 41.6 | 61.9 | 45.6 | 37.4 | 58.8 | 40.2 | 76.5 | 71.9 | 37.8 |
| *Ours* (SlotCon) | 100 | ✗ | ✗ | 41.4 | 61.6 | 45.6 | 37.2 | 58.5 | 39.9 | 75.4 | 73.1 | 38.6 |
| *Ours* (SlotCon) | 200 | ✗ | ✗ | **41.8** | **62.2** | 45.7 | **37.8** | **59.1** | **40.7** | 76.3 | **75.0** | **38.8** |

### 4.4 Unsupervised semantic segmentation results

Given our approach's consistent improvement in representation learning, we further analysis of how well our semantic grouping component can parse scenes quantitatively and qualitatively. It should be

---

[2]PixPro aggregates the global context with self-attention [63], so each pixel can also be viewed as an object-level embedding.

Table 5: Main results in COCO-Stuff unsupervised semantic segmentation.

| Method | mIoU | pAcc |
|---|---|---|
| MaskContrast [62] | 8.86 | 23.03 |
| PiCIE + H. [39] | 14.36 | 49.99 |
| SegDiscover [37] | 14.34 | **56.53** |
| *Ours* (SlotCon) | **18.26** | 42.36 |

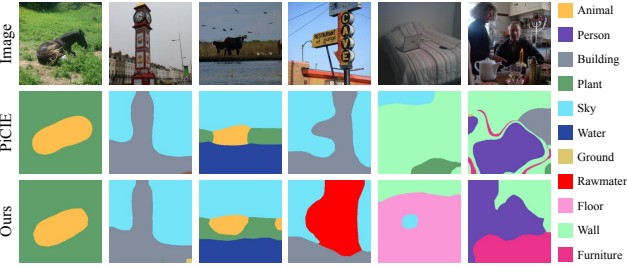

clarified that this is just to help understand the learned prototypes, and not to propose a new SOTA. Unlike current SSL approaches that exhaustively enumerate the massive object proposals and report the best score [7, 35], we follow the common practice of unsupervised semantic segmentation [40, 39] to match the predicted results with the ground-truth using the Hungarian algorithm [43], where each ground-truth label is assigned to a prototype mutual-exclusively. For fair comparisons, the model used for evaluation is trained with $K = 27$ to match the number of categories of COCO-Stuff. As shown in Table 5, our method can surpass the previous works PiCIE [39] and SegDiscover [37] with 4 points higher mIoU. Meanwhile, the pAcc is lower since we train the model with a lower resolution ($7 \times 7$ v.s. $80 \times 80$ feature map). Besides Table 5, we also depict the visualization results, in which our method distinguishes confusing objects apart (4th column) and successfully localizes small objects (5th column). Since we only need to separate pixels with different semantics within the same image, the errors in category prediction can be ignored.

## 4.5 Ablation study

Table 6: **Ablation studies with COCO 800 epochs pre-training.** We show the $AP^b$ on COCO objection detection and mIoU on Cityscapes, PASCAL VOC, and ADE20K semantic segmentation. The default options are marked with a gray background.

(a) **Number of prototypes**

| $K$ | COCO | City | VOC | ADE |
|---|---|---|---|---|
| 128 | 40.7 | **76.4** | **71.9** | 38.5 |
| 256 | **41.0** | 76.2 | 71.6 | 39.0 |
| 512 | 40.9 | 75.6 | 71.6 | 38.9 |
| 1024 | 40.7 | 75.8 | 70.9 | **39.1** |

(b) **Loss balancing**

| $\lambda_g$ | COCO | City | VOC | ADE |
|---|---|---|---|---|
| 0.3 | **41.0** | 76.1 | **72.1** | 37.9 |
| 0.5 | **41.0** | **76.2** | 71.6 | **39.0** |
| 0.7 | 40.5 | 75.2 | 71.5 | 38.4 |
| 1.0 | 40.4 | 74.2 | 70.1 | 38.6 |

(c) **Teacher temperature**

| $\tau_t$ | COCO | City | VOC | ADE |
|---|---|---|---|---|
| 0.04 | 40.4 | 75.5 | 70.2 | 37.9 |
| 0.07 | **41.0** | **76.2** | **71.6** | **39.0** |

**Number of prototypes $K$.** Table 6a ablates the number of prototypes, we observe that the most suitable $K$ for COCO detection is 256, which is close to its real semantic class number 172 (thing + stuff) [4]. Besides, the performance on Cityscapes and PASCAL VOC have a consistent tendency to drop, while the performance on ADE20K is consistently good if $K$ is big enough. We hypothesize that a suitable $K$ can encourage learning data-specific semantic features, which are only helpful when the pre-training and downstream data are alike (from COCO to COCO); increasing $K$ produces fine-grained features that may lack discriminability in semantics but hold better transferability to ADE20K that require fine-grained segmentation [14].

**Loss balancing weight $\lambda_g$.** Table 6b ablates the balancing between the semantic grouping loss and the group-level contrastive loss, where the best balance is achieved with both losses treated equally. It is notable that when $\lambda_g = 1.0$, only the semantic grouping loss is applied, and the performance drops considerably, indicating the importance of our group-level contrastive loss for learning good representations.

**Teacher temperature $\tau_t$.** Table 6c ablates the teacher model's temperature parameter, indicating that a softer teacher distribution with $\tau_t = 0.07$ helps achieve better performance.

## 4.6 Probing the prototypes

Finally, we analyze whether the prototypes learn semantic meanings by visualizing their nearest neighbors in COCO val2017 split. We first perform semantic grouping on each image to split them into non-overlapping groups (segments), then pool each group to a feature vector, and retrieve the

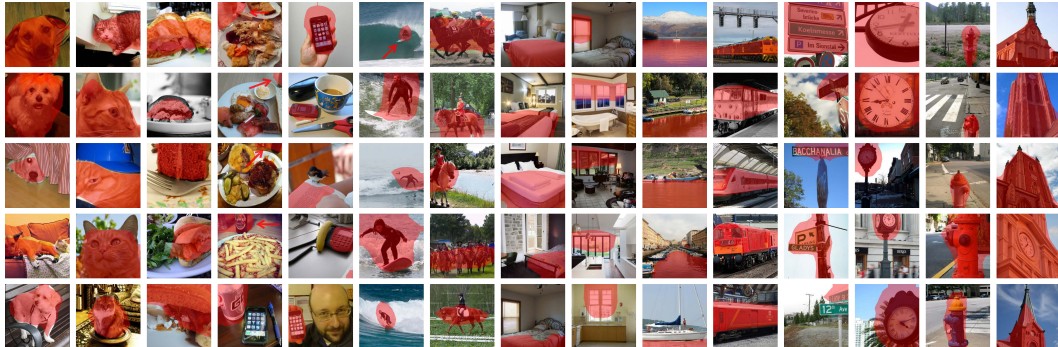

Figure 3: **Examples of visual concepts discovered by SlotCon from the COCO** `val2017` **split.** Each column shows the top 5 segments retrieved with the same prototype, marked with reddish masks or arrows. Our method can discover visual concepts across various scenarios and semantic granularities regardless of small object size and occlusion. (*best viewed in color*)

top 5 nearest-neighbor segments for each prototype according to cosine similarity. As shown in Figure 3, the prototypes well bind to semantic meanings that cover a wide range of scenarios and semantic granularities from animals, foods, and sports, to furniture, buildings, etc., localizing them well regardless of small object size and occlusion; and notably, *without any human annotation*.

## 5 Discussion on the emergence of objectness

It should be clarified that the discovered objectiveness is at the semantic level, and object instances with identical semantics can be indistinguishable. Concerning the emergence of objectness, we impose geometric-covariance and photometric-invariance as guiding cues which force the model to decompose a large complex dataset into a small number of clusters through optimizing feature space and cluster centers. The emergence of objects/parts that are compositional and thus occupy a reasonable number of prototypes, can be viewed as a natural consequence under such conditions. Concerning granularity, our intuition is that the prototype number and the dataset distribution generate a bottleneck for the granularity of groups. For example, in the COCO dataset, with 256 prototypes, the model finds that splitting animals into cats, dots, elephants, *etc.*, is enough and won't further separate them. In contrast, for humans (the most occupying category of COCO), as shown in the appendix, the model discovers not only human parts (Figure 7) but also different types of activities (Figure 8), indicating that parts are more helpful in this scenario and deserve more prototypes.

## 6 Conclusion

This work presents a unified framework for joint semantic grouping and representation learning from unlabeled scene-centric images. The semantic grouping is performed by assigning pixels to a set of learnable prototypes, which can adapt to each sample by attentive pooling over the feature map and form new slots. Based on the learned data-dependent slots, a contrastive objective is employed for representation learning, enhancing features' discriminability and facilitating the grouping of semantically coherent pixels together. By simultaneously optimizing the two coupled objectives of semantic grouping and contrastive learning, the proposed approach bypasses the disadvantages of handcrafted priors and can learn object/group-level representations from scene-centric images. Experiments show the proposed approach effectively decomposes complex scenes into semantic groups for feature learning and significantly facilitates downstream tasks, including object detection, instance segmentation, and semantic segmentation.

## Acknowledgments

This work has been supported by Hong Kong Research Grant Council - Early Career Scheme (Grant No. 27209621), HKU Startup Fund, and HKU Seed Fund for Basic Research. The authors acknowledge SmartMore, LunarAI, and MEGVII for partial computing support.

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
