# Supplementary Material for
# Self-Supervised Visual Representation Learning with Semantic Grouping

**Xin Wen**[1]    **Bingchen Zhao**[2,3]    **Anlin Zheng**[1,4]    **Xiangyu Zhang**[4]    **Xiaojuan Qi**[1]

[1]University of Hong Kong    [2]University of Edinburgh    [3]LunarAI    [4]MEGVII Technology

{wenxin, xjqi}@eee.hku.hk    zhaobc.gm@gmail.com
{zhenganlin, zhangxiangyu}@megvii.com

## Contents

# A   Additional implementation details

## A.1   Inverse augmentation

The inverse augmentation process aims to recover the original pixel locations and ensure the two feature maps produced from two augmented views are spatially aligned after inverse augmentation. There are two operations in our data augmentation pipeline that changes the scale or layout of the image, *i.e.*, random resized crop and random horizontal flip. Since we already know the spatial coordinates where each view is cropped from, we can map the coordinates to the corresponding feature maps and cut the rectangular part from the feature map where the two sets of coordinates intersect. This is followed by a resize operation to recover the intersect part to the original size (*e.g.*, $7 \times 7$ for a $224 \times 224$ input). In implementation, we achieve this through RoIAlign [10]. If the horizontal flip operation is applied to produce the view, we also use a horizontal flip operation after RoIAlign to recover the original spatial layout. After the inverse augmentation, each pixel in the two feature maps is spatial-aligned, making applying the per-pixel cross-entropy loss easy.

## A.2   Transfer learning

### A.2.1   Object detection and instance segmentation

We train a Mask R-CNN [10] model with R50-FPN backbone [17] implemented with the open-source project `Detectron2` [24], following the same fine-tuning setup with [22, 26, 25]. Specifically, we use a batch size of 16, and fine-tune for 90k iterations (standard $1\times$ schedule) with batch normalization layers synchronized. The learning rate is initialized as 0.02 with a linear warm-up for 1000 iterations, and decayed by 0.1 at 60k and 80k iterations. The image scale is $[640, 800]$ during training and 800 at inference. We fine-tune all layers end-to-end on COCO [18] `train2017` set with the standard $1\times$ schedule and report AP, $AP_{50}$, $AP_{75}$ on the `val2017` set.

### A.2.2   Semantic segmentation

**Cityscapes and PASCAL VOC.**   We strictly follow [9] for transfer learning on these two datasets. Specifically, we use the same fully-convolutional network (FCN)-based [20] architecture as [9]. The backbone consists of the convolutional layers in ResNet-50, in which the $3 \times 3$ convolutions in conv5 blocks have dilation 2 and stride 1. This is followed by two extra $3 \times 3$ convolutions of 256 channels (dilation set to 6), with batch normalization and ReLU activations, and then a $1 \times 1$ convolution for per-pixel classification. The total stride is 16 (FCN-16s [20]).

Training is performed with random scaling (by a ratio in $[0.5, 2.0]$), cropping, and horizontal flipping. The crop size is 513 on PASCAL VOC [6] and 769 on Cityscapes [4], and inference is performed on the original image size. We train with batch size 16 and weight decay 0.0001. The learning rate is 0.003 on VOC and is 0.01 on Cityscapes (multiplied by 0.1 at the 70th and 90th percentile of training). For PASCAL VOC, we fine-tune the model on `train_aug2012` set for 30k iterations and report the mean intersection over union (mIoU) on the `val2012` set. For Cityscapes, we fine-tune on the `train_fine` set for 90k iterations and evaluate it on the `val_fine` set.

**ADE20K.**   For ADE20K [28], we train with a FCN-8s [20] model on the `train` set and evaluate on the `val` set, and the optimization specifics follows the standard 80k iterations schedule of `MMSegmentation` [3]. Specifically, we fine-tune for 80k iterations with stochastic gradient descent, with a batch size of 16 and weight decay of 0.0005. The learning rate is 0.01 and decays following the poly schedule with power of 0.9 and min_lr of 0.0001.

## A.3   Unsupervised semantic segmentation

**Experiment setting.**   We follow the common practice in this field [14, 13, 12] to use a modified version of COCO-Stuff [1], where the labels are merged into 27 categories (15 "stuff" categories and 12 "thing" categories). We perform inference with resolution 320 and number of prototypes 27 following the common practice, and evaluate on mIoU and pixel accuracy (pAcc).

**Inference details.**   Intuitively, each prototype can be viewed as the cluster center of a semantic class. Therefore, we simply adopt the prototypes $\mathcal{S} \in \mathbb{R}^{K \times D}$ as a $1 \times 1$ convolution layer for per-pixel

classification, and predict the prototypical correspondence of each pixel with the argmax operation.

$$\hat{y} = \underset{K}{\mathrm{argmax}} \left( \mathrm{resize} \left( \overline{\boldsymbol{z}} \cdot \overline{\mathcal{S}}^{\top} \right) \right) \in \mathbb{Z}^{H' \times W'} , \tag{1}$$

where the resize operation denotes bi-linear interpolation on the logits to the size of the image ($320 \times 320$ in this case). To match the prototypes with the ground truth clusters, we follow the standard protocol [14, 13] of finding the best one-to-one permutation mapping using Hungarian-matching [16]. Then the pAcc and mIoU are calculated according to the common practice [13]. During inference, we only take the teacher model parameterized by $\xi$.

### A.4 Visual concept discovery

Simply speaking, the visual concept discovery is similar to the semantic segment retrieval task [21], except that the queries are prototypes rather than segments. Specifically, we adopt the COCO val2017 set, consisting of 5k images, and the default model trained on COCO with the number of prototypes $K = 256$. Each image is first resized to 256 pixels along the shorter side, after which a $224 \times 224$ center crop is applied. We then follow Eq. 1 to assign a prototype index to each pixel; thus, each image is split into a set of groups, such that the pixels within each group hold the same prototypical assignments. We rephrase the groups as *segments*, and compute the feature vector for each segment by average pooling. Then for each prototype, we calculate the cosine similarity between it and all segments in the dataset assigned to this prototype and retrieve those with top-$k$ high similarity scores.

### A.5 Re-implementing related works

Some current works may differ in implementation details for downstream tasks (*e.g.*, SoCo [23] uses different hyper-parameters for COCO object detection and instance segmentation, DetCon [11] uses different hyper-parameters for semantic segmentation, and DenseCL [22] adopts different network architectures for semantic segmentation). For a fair comparison, we re-produced the transfer learning results with a unified setting with the official checkpoints and re-implement the pre-training with the official code if needed.

## B Additional transfer learning results

### B.1 Longer COCO schedule

Table 1: **Additional transfer learning results with COCO 800 epochs pre-training.** We report the results in COCO object detection and COCO instance segmentation with both $1\times$ and $2\times$ schedules.

| Method | Transfer learning schedule | COCO detection | | | COCO segmentation | | |
|---|---|---|---|---|---|---|---|
| | | $\mathrm{AP^b}$ | $\mathrm{AP^b_{50}}$ | $\mathrm{AP^b_{75}}$ | $\mathrm{AP^m}$ | $\mathrm{AP^m_{50}}$ | $\mathrm{AP^m_{75}}$ |
| SlotCon | $1\times$ (90k iterations) | 41.0 | 61.1 | 45.0 | 37.0 | 58.3 | 39.8 |
| SlotCon | $2\times$ (180k iterations) | 42.6 | 62.7 | 46.2 | 38.2 | 59.6 | 41.0 |

In Table 1, we further provide the downstream results of SlotCon in COCO object detection and instance segmentation with a longer transfer learning schedule ($2\times$). Compared with the results with the $1\times$ schedule, it shows significant improvements in all metrics.

### B.2 Pre-training with autonomous driving data

In Table 2, we show the results with BDD100K [27] pre-training and evaluated on Cityscapes semantic segmentation. The model is trained on BDD100K for $800$ epochs with $64$ prototypes. The result is notably weaker than its COCO counterpart, yet still surpasses MoCo v2 pre-trained on COCO. The BDD100K dataset is indeed challenging for pre-training as its images are less discriminative, and the task of pre-training on autonomous driving data is a valuable direction for future explorations.

Table 2: Transfer learning results with BDD100K pre-training.

| Pre-train Data | Method | Cityscapes mIoU |
|---|---|---|
| - | Random init. | 65.3 |
| COCO | MoCo v2 | 73.8 |
| COCO | SlotCon | 76.2 |
| BDD100K | SlotCon | 73.9 |

## C   Additional ablation studies

Table 3: **Ablation studies with COCO 800 epochs pre-training.** We show the $AP^b$ on COCO objection detection and mIoU on Cityscapes, PASCAL VOC, and ADE20K semantic segmentation. The default options are marked with a gray background.

(a) **Batch size**

| B | COCO | City | VOC | ADE |
|---|---|---|---|---|
| 256 | 40.6 | 75.9 | 70.9 | 38.1 |
| 512 | **41.0** | **76.2** | 71.6 | **39.0** |
| 1024 | 40.7 | 75.7 | **71.8** | 38.6 |

(b) **Type of group-level loss**

| Loss | COCO | City | VOC | ADE |
|---|---|---|---|---|
| Reg. | 40.7 | 75.9 | 71.0 | **39.0** |
| Ctr. | **41.0** | **76.2** | **71.6** | **39.0** |

(c) **Where to apply** invaug**?**

| Align | COCO | City | VOC | ADE |
|---|---|---|---|---|
| Proj. | 40.9 | 75.7 | 71.4 | 38.0 |
| Asgn. | **41.0** | **76.2** | **71.6** | **39.0** |

(d) **Batch size and image-level objective**

| B | $\mathcal{L}_{Image}$ | COCO $AP^b$ | COCO $AP^m$ |
|---|---|---|---|
| 512 | ✗ | 41.0 | **37.0** |
| 512 | ✓ | 40.8 | 36.8 |
| 1024 | ✗ | 40.7 | 36.7 |
| 1024 | ✓ | **41.1** | **37.0** |

(e) **Geometric augmentations**

| Method | Geometric aug. | VOC mIoU |
|---|---|---|
| Random init. | - | 39.5 |
| SlotCon | ✓ | **71.6** |
| SlotCon | ✗ | 62.6 |

In Table 3, we provide the results of further ablation studies in batch size, the type of group-level contrastive loss, the place to apply inverse augmentation, the influence of an image-level learning objective, and the importance of geometric augmentations. We discuss them as follows:

**Batch size.**   Table 3a shows the most suitable batch size for our method is 512. Increasing it to 1024 does not result in better performance. We argue that the real slot-level batch size is actually bigger than 512, and should be multiplied by the number of pixels (49) or slots (∼8) per image for the grouping loss and the group-level contrastive loss, respectively. Considering the mismatch in the batch size scale of the two loss functions, the learning rate might should be further tuned to work with larger batches [7].

**Type of group-level loss.**   Table 3b shows that both the BYOL [8]-style regression loss and the contrastive loss are helpful to learning transferable features, and the results with the contrastive loss are especially higher for object detection in COCO. This may indicate that the contrastive loss, which better pushes negative samples apart, is beneficial for object detection, in which the ability to tell confusing objects apart is also critical.

**Place to apply inverse augmentation.**   Table 3c ablates whether to apply the inverse augmentation operation on the dense projections or the grouping assignments, and shows the latter is better. This can keep the non-overlapping features for the group-level contrastive loss and utilize more information.

**Image-level learning objective.**   Table 3d ablates the effect of an auxiliary image-level learning objective. We add a MoCo v3 [2] style image-level contrastive learning loss to SlotCon (COCO 800 epoch pre-training setting), and the experiment results on COCO show that it depends on the batch size. With a smaller batch size of 512, the detection AP drops by 0.2 points, while with a higher batch size of 1024, the detection AP rises by 0.4 points. Our explanation is that the image-level objective is more sensitive to the batch size, and it requires a larger batch size to learn holistic representations that are complementary to the object-level objective.

**Geometric augmentations.** Table 3e ablates the effect of the geometric augmentations. Our main finding is that geometric augmentations are necessary to learn object-centric representations in our setting. We train SlotCon on COCO for 800 epochs with two identical crops applied for each image, thus only the photometric invariance is adopted as the supervision. We then visualized the slots the same way as Figure 3 in the main paper, and found that almost none of the slots can bind to a meaningful semantic. Most of them attend to a similar-shaped region that locates at the same position across different images while holding diverging semantics. And some of them learn textures like animal fur, cloudy sky, snowland, or leaves. A fast evaluation on PASCAL VOC semantic segmentation shows a significant performance drop, yet the feature is still notably better than random initialization.

## D   Statistics about the binary indicator

**How many slots are active on average for each image?** It depends on the number of categories/semantics per image. As shown in Figure 1, seven slots are active on average for one image after convergence.

**How often is one slot active over the whole dataset?** It depends on the category/semantic distribution of the dataset, as the slots are roughly bound to real-world semantic categories. We studied the activeness of the slots over the COCO `val2017` set that contains 5000 images, and found that 40 out of the 256 slots are dead, and not active to any image. The activeness of the remaining slots follows a long-tailed distribution. The top-5 active slots correspond to tree (376), sky (337), streetside car (327), building exterior wall (313), and indoor wall (307); and the bottom-5 active slots correspond to skateboarder (44), grassland (45), train (56), luggage (57), and airplane (57).

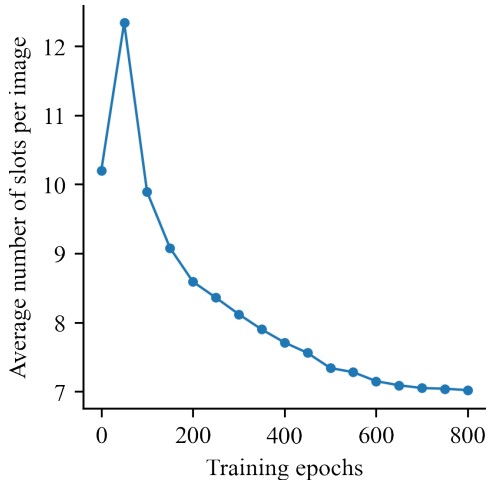

Figure 1: Average number of active slots per image during training on COCO.

**How many terms are excluded in Eq. 6 in the main paper?** Given the slot number as 256, if all slots are active for one image and the two crops overlap well, there should be 256 positive pairs. We studied a well-converged model and found that, on average, only around 3.6 pairs are active, so around 252.4 terms are excluded. It is reasonable considering that around 7 slots are active on one image, and the overlapping area between the two crops can be small. Considering a total batch size of 512, the number of excluded negative samples should be around $512 \times (256 - 7) = 127488$.

## E   Computational costs

In Table 3, we give a direct computational cost comparison between SlotCon and two previous works. The experiments are conducted on the same machine with 8 NVIDIA GeForce RTX 3090 GPUs. Both PixPro and SlotCon adopt a batch size of 1024 and have amp turned on, and DenseCL adopts a batch size of 256 by default. The training time of DenseCL might be higher than optimal as we failed to install apex.

Table 3: Computational cost evaluation.

| Method | Time/epoch | Memory/GPU |
|---|---|---|
| DenseCL [22] | $2'46''$ | 7.9 GB |
| PixPro [26] | $2'19''$ | 15.1 GB |
| SlotCon | $2'23''$ | 16.0 GB |

## F   Additional qualitative results

**Unsupervised semantic segmentation.** In Figure 2, 3, 4, we provide the visualization of more results in COCO-Stuff unsupervised semantic segmentation. Compared with PiCIE [13], our method's overall successes in distinguishing confusing objects apart and localizing small objects.

**Visual concept discovery on COCO.**    In Figure 5, 6, we show more results of visual concepts discovered by our model from COCO, which cover a wide range of natural scenes. We further show that the model tends to categorize person-related concepts into fine-grained clusters. For example, in Figure 7, we show that it also groups segments according to the part of the human body; and in Figure 8, we show that it also groups person segments by the sport they are playing. We hypothesize that persons are too common in COCO, and the model finds that allocating more prototypes to learn person-related concepts can better help optimize the grouping loss.

**Visual concept discovery on ImageNet.**    In Figure 9, 10, we also provide examples of visual concepts discovered by our method from ImageNet. Due to the scale of ImageNet, it is hard to compute the segments for all the images. As ImageNet is basically single-object-centric, we simply treat each image as a single segment to save computation for nearest-neighbor searching. The visualization verifies the compatibility of our method with object-centric data.

## G    Limitations and negative social impacts

**Grouping precision.**    Since we directly learn a set of semantic prototypes with a quite low-resolution feature map ($32\times$ downsample) and do not have any supervision for precise object boundaries, it is hard for our model to perform detailed semantic grouping and cases are that many foreground instances are segmented with over-confidence. Using post-processing through iterative refinements such as CRF [15] or pre-compute visual primitives (super-pixels) on the raw image [12] may improve the result, but they are out of the scope of this work. Besides, modern object discovery techniques such as Slot Attention [19] that incorporate attention mechanism and iterative refinement may also help learn better semantic groups; we leave this for future work.

**Training cost.**    As all self-supervised learning methods do, our approach also needs to pre-train with multiple GPU devices for a long time, which may increase carbon emissions. However, for one thing, the pre-training only needs to be done once and can help reduce the training time of multiple downstream tasks; for another, our method can learn relatively good representations with shorter training time, *e.g.*, our method pre-trained on ImageNet for $100$ epochs achieves compatible performance with PixPro [26] pre-trained for $400$ epochs in COCO objection detection ($AP^b = 41.4$).

**The data can be unreliable.**    With reduced human priors, our method learns to discover objects/semantics from large-scale natural images. However, totally relying on the data may lead to "bad" biases. In our experiments on scene-centric data, we noticed that the model allocates more prototypes to human-related concepts (as humans occur most frequently in COCO), while many other kinds of animals only have one prototype (see Figure 6, 8, 7). When pre-training on a more long-tailed and less discriminative scenario (*e.g.*, autonomous driving data, detailed in Sec B.2), the data can lead to highly biased prototypes and representations, and harm downstream performance. Injecting human priors to guide the criterion for objects/semantics could be a promising direction.

**Semantic rather than object.**    Our method performs semantic grouping; this means that objects with identical semantics can be indistinguishable. Therefore, the contrastive learning objective (Eq. 7 in the main paper) does not contrast between objects in the same image with the same semantic (*e.g.*, two elephants in one image). However, the success of self-supervised learning has shown that discriminating instances with identical category labels can further boost representation quality. The transition from semantic to objectness is to be explored.

## H    License of used datasets

All the datasets used in this paper are permitted for research use. The terms of access to the images of COCO [18] and ImageNet [5] allow the use for non-commercial research and educational purposes. Besides, the annotations of COCO [18] and COCO-Stuff [1] follow the Creative Commons Attribution 4.0 License, also allowing for research purposes.

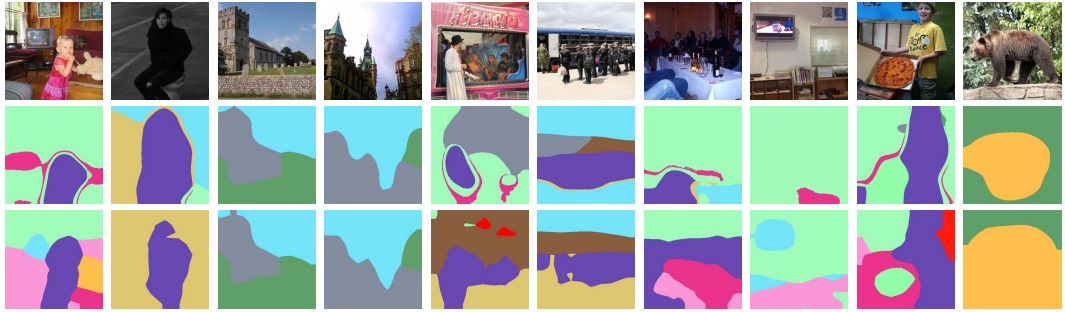

Figure 2: Additional results in COCO-Stuff [1] unsupervised semantic segmentation. Each row from top to down: Image, PiCIE [13], Ours. Overall, our method successfully distinguishes confusing objects and localizes small objects. (*best viewed in color*)

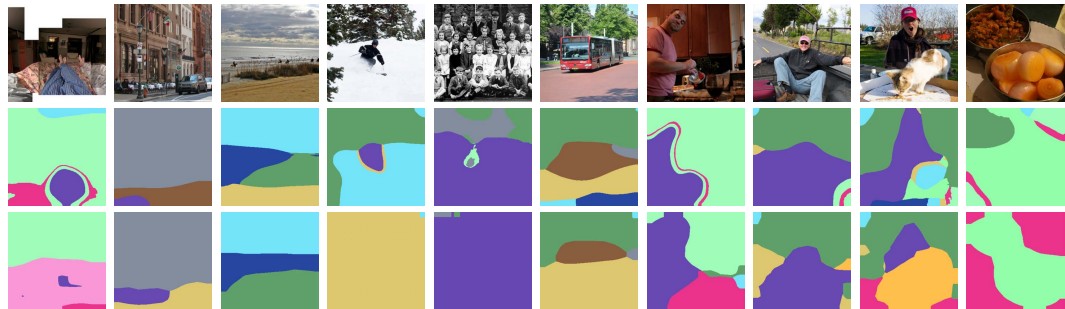

Figure 3: Additional results in COCO-Stuff [1] unsupervised semantic segmentation. Each row from top to down: Image, PiCIE [13], Ours. Overall, our method successfully distinguishes confusing objects and localizes small objects. (*best viewed in color*)

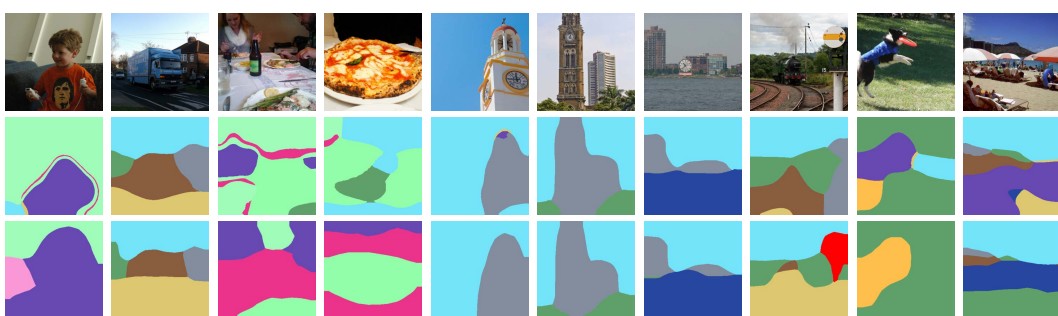

Figure 4: Additional results in COCO-Stuff [1] unsupervised semantic segmentation. Each row from top to down: Image, PiCIE [13], Ours. Overall, our method successfully distinguishes confusing objects and localizes small objects. (*best viewed in color*)

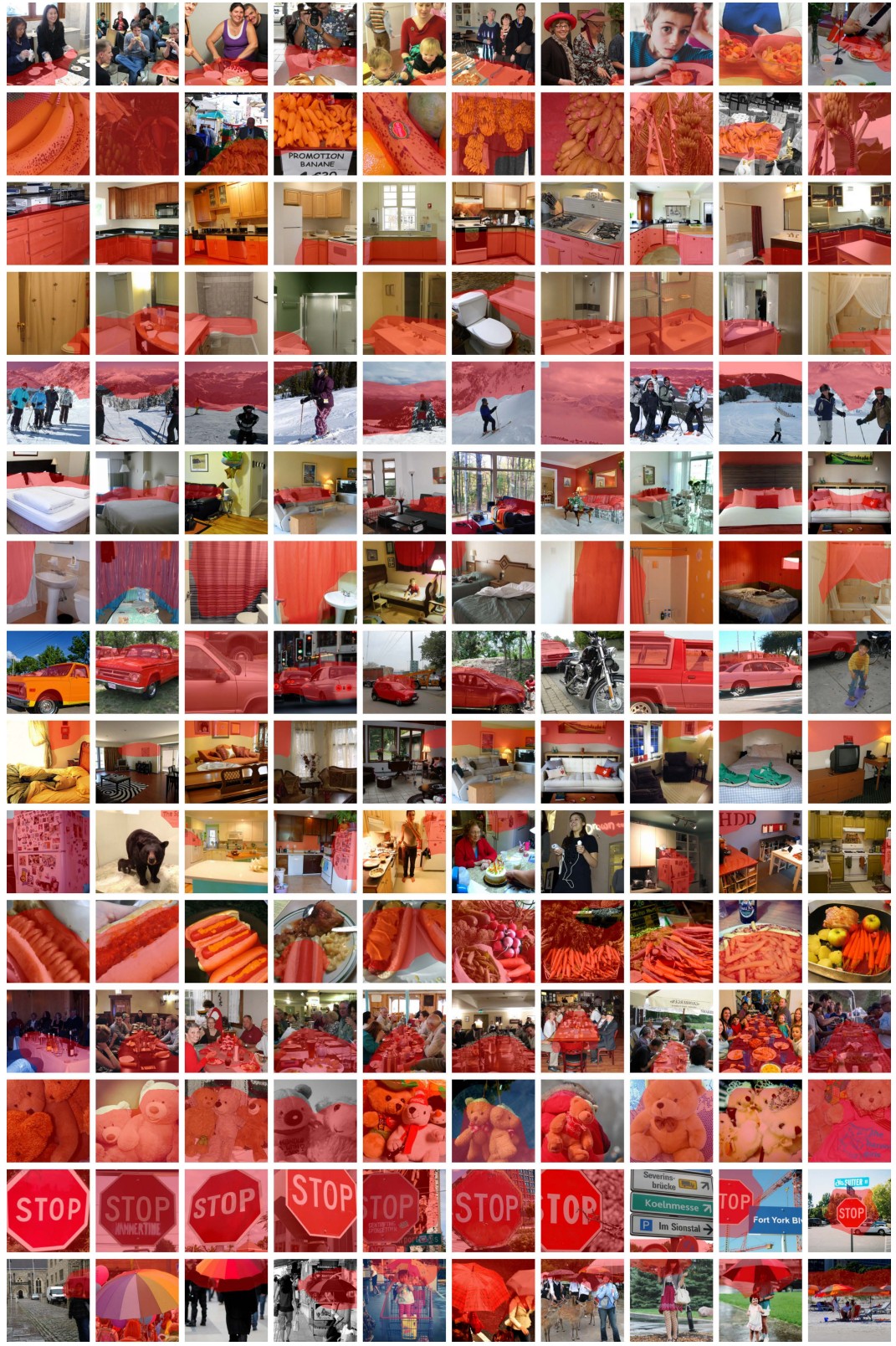

Figure 5: Additional examples of visual concepts discovered by our method from the COCO `val2017` split. Each row shows the top 10 segments retrieved with the same prototype, marked with red masks. (*best viewed in color*)

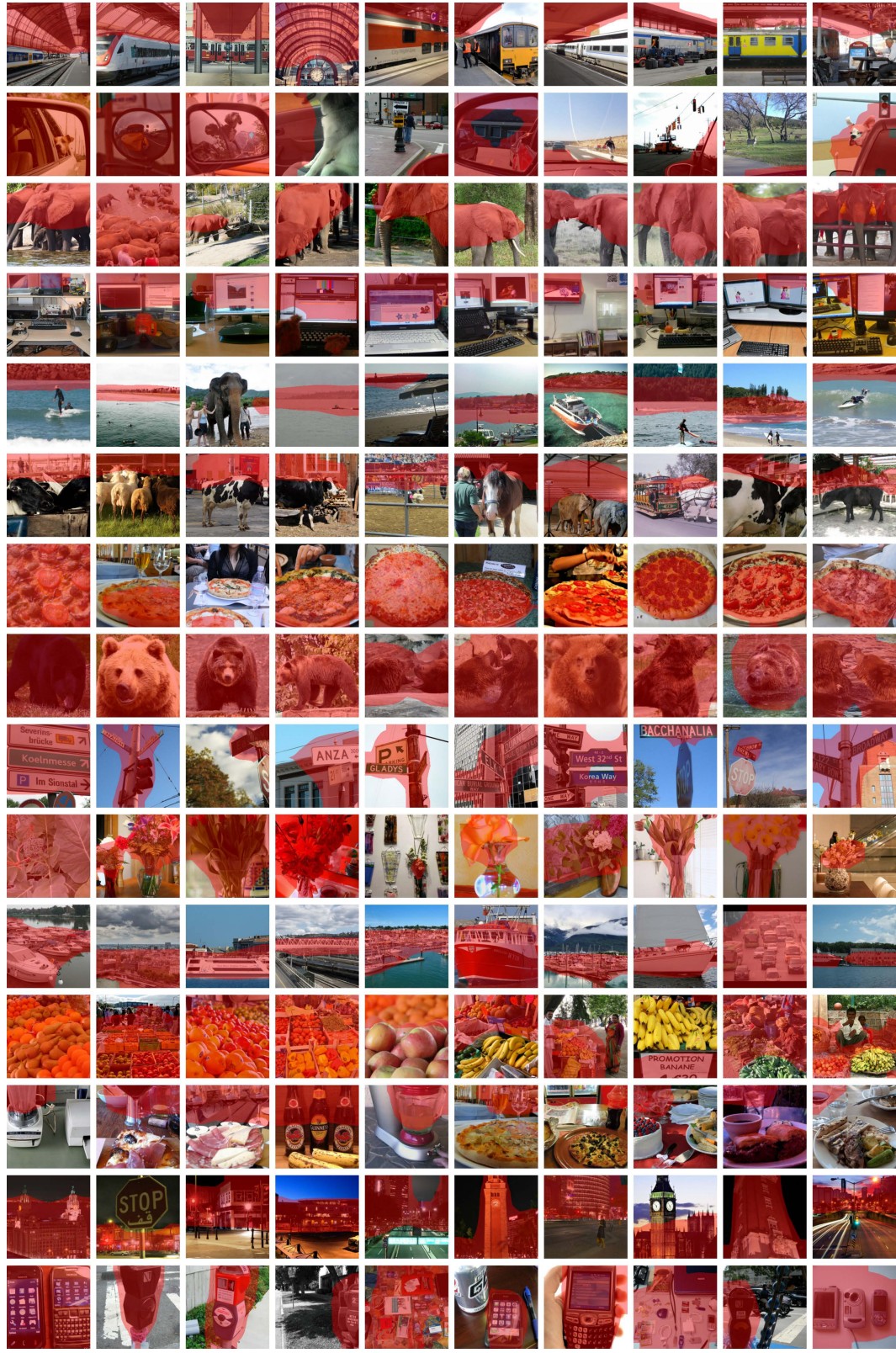

Figure 6: Additional examples of visual concepts discovered by our method from the COCO `val2017` split. Each row shows the top 10 segments retrieved with the same prototype, marked with red masks. (*best viewed in color*)

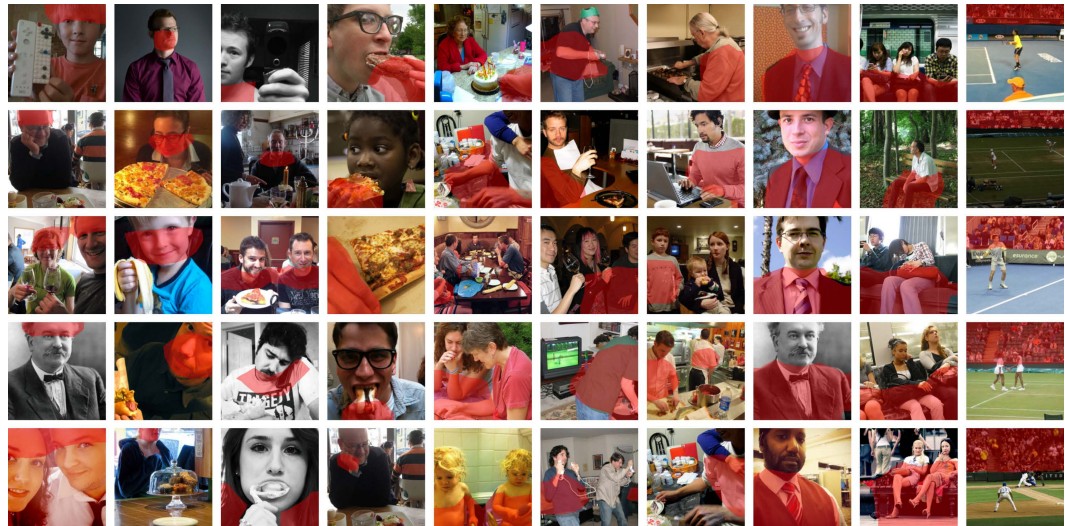

Figure 7: Additional examples of visual concepts discovered by our method from the COCO `val2017` split. Each column shows the top 5 segments retrieved with the same prototype, marked with red masks. The model tends to group person-related segments into fine-grained clusters. This figure shows those related to different granularities of the human body, including the forehead, face, shoulder & neck, hand, arm, elbow, chest, leg, and crowd. (*best viewed in color*)

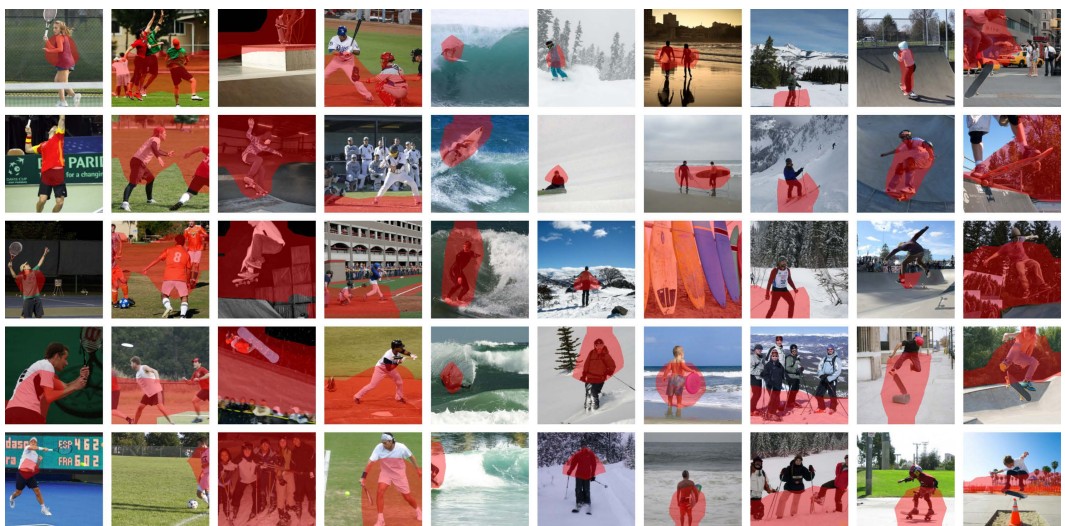

Figure 8: Additional examples of visual concepts discovered by our method from the COCO `val2017` split. Each column shows the top 5 segments retrieved with the same prototype, marked with red masks. The model tends to group person-related segments into fine-grained clusters. This figure shows those related to different types of sports, including tennis, football, skateboarding, baseball, surfing, and skiing. (*best viewed in color*)

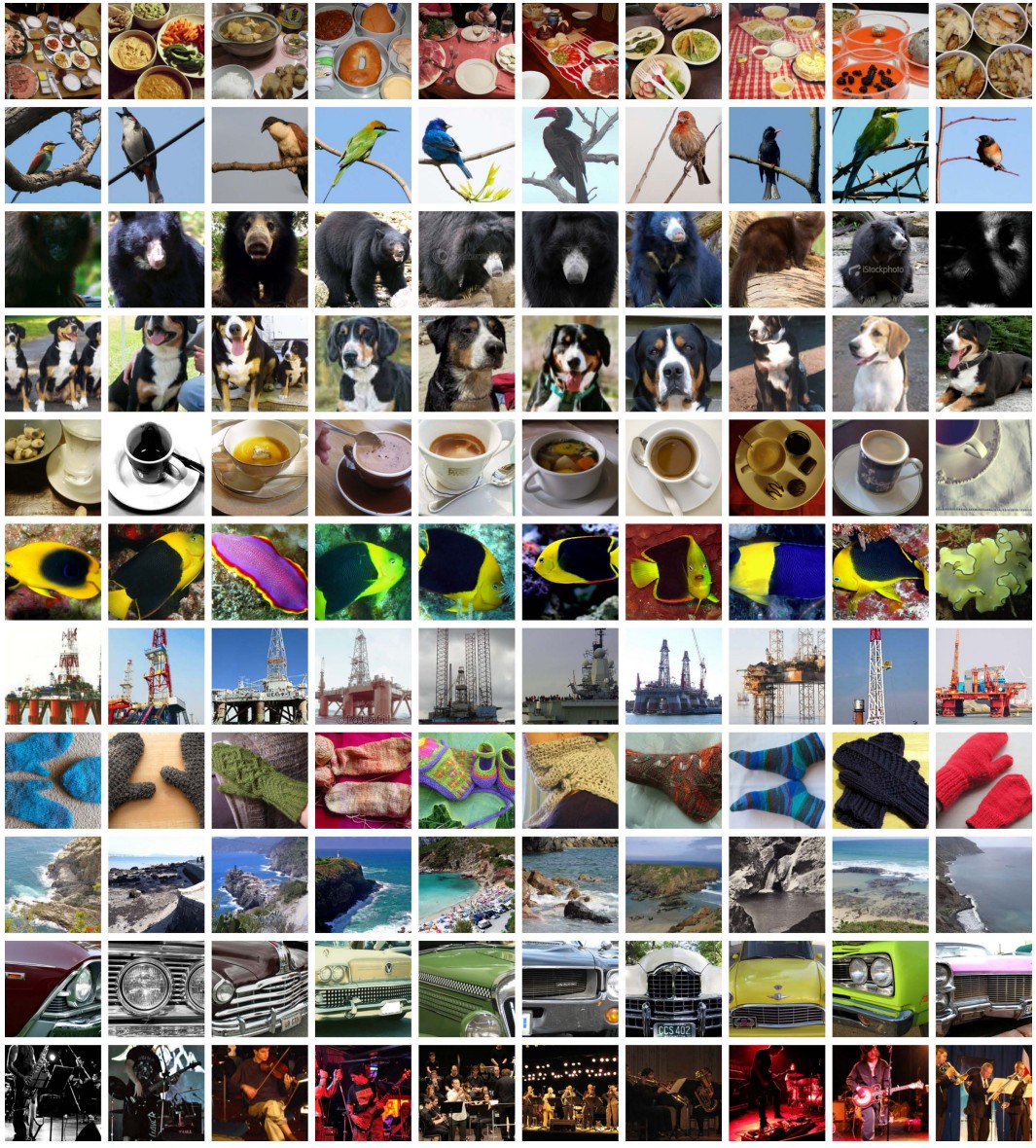

Figure 9: Additional examples of visual concepts discovered by our method from ImageNet [5]. Each row shows the top 10 images retrieved with the same prototype. Due to the scale of ImageNet, it is hard to compute the segments for all the images. As ImageNet is basically single-object-centric, we simply treat each image as a single segment to save computation for nearest-neighbor searching. The result verifies our method's compatibility with object-centric data. (*best viewed in color*)

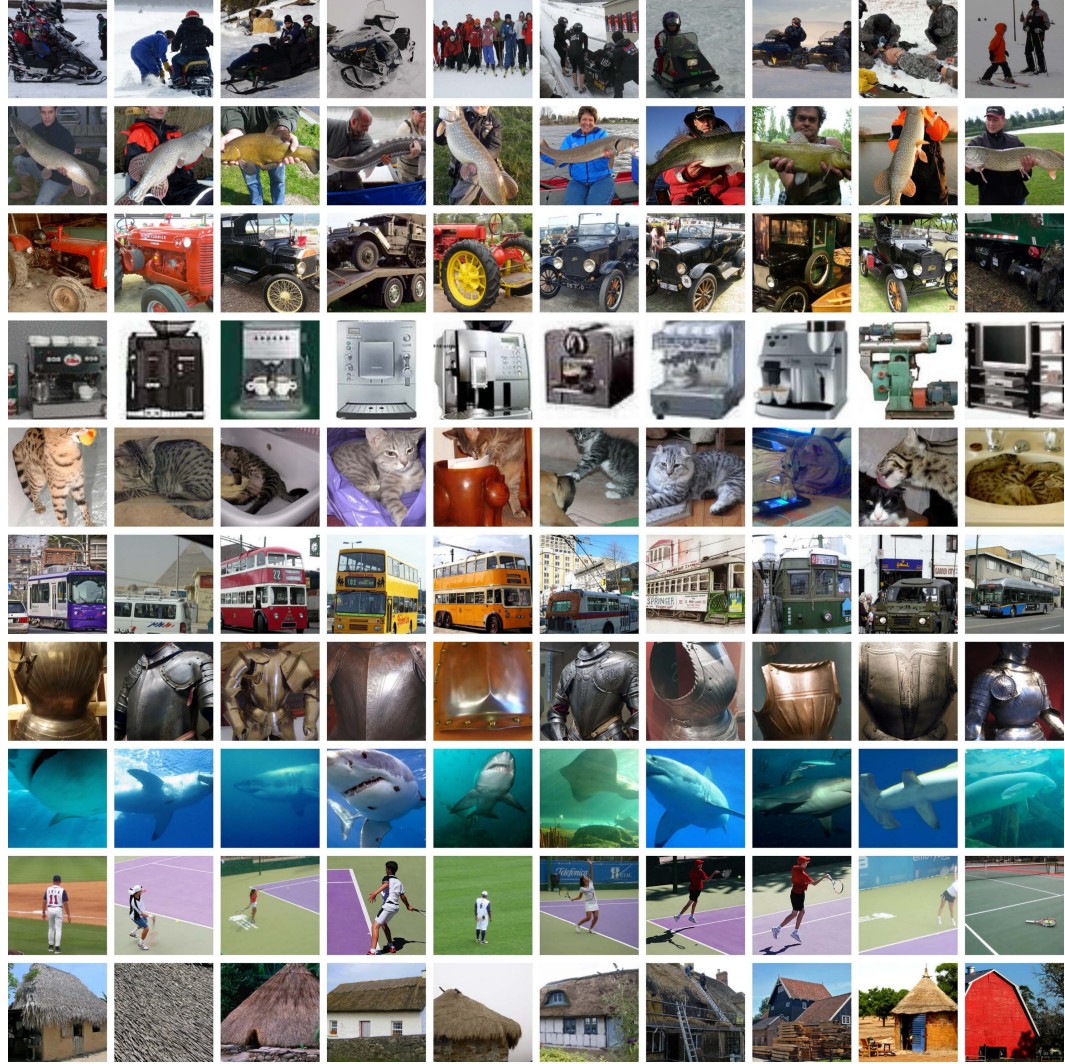

Figure 10: Additional examples of visual concepts discovered by our method from ImageNet [5]. Each row shows the top 10 images retrieved with the same prototype. Due to the scale of ImageNet, it is hard to compute the segments for all the images. As ImageNet is basically single-object-centric, we simply treat each image as a single segment to save computation for nearest-neighbor searching. The result verifies our method's compatibility with object-centric data. (*best viewed in color*)