# OpenReview forum: "Self-Supervised Visual Representation Learning with Semantic Grouping"
_NeurIPS.cc/2022/Conference — NeurIPS 2022 Accept_

### Official Review · Reviewer_NYjs · 2022-07-08

**Rating:** 6
**Confidence:** 4
**Soundness:** 3 good
**Presentation:** 3 good
**Contribution:** 3 good

**Summary:**

The authors design a method to learn visual representations in a self-supervised fashion from scene-centric data. The work is built upon the choice of exploiting intrinsic scene-centric characteristics of the data, and contrasting with existing literature that exploits object-related priors to guide the learning process.

The method includes a pixel-level semantic grouping learning mechanism based on learning a set of prototypes to which the pixels are assigned. The set of prototypes can adapt to each sample by an adaptive pooling mechanism.

**Questions:**

I would revisit the statements about the use of prior knowledge as a guidance to the training process, and relate this work better with existing streams of work that show successful use of priors both for performance improvement and data-efficiency (e.g. in the VIPriors workshop series).

The results to not clearly show superiority of the proposed approach, lacking a full support of some statements in the introduction. Neither the results show complementarity, in the way they are presented. An error and overlap prediction analysis wrt other methods would contribute to a better understanding of the cases in which one method is better than another. Also, marginal improvements might be checked with statistical tests of significance. If not done, these marginal improvements are bound to be interpreted as a result of randomness.

**Limitations:**

The authors do not state limitations explicitly. They could discuss more extensively in which cases this approach would be better than others and in which cases not. From the results it is not clear why and when this method would be better to be used.
I do not find direct implications as potentially negative societal impact. It is a general approach to learn CV models, and its uses are very broad.

**Strengths And Weaknesses:**

*Strenghts*
- novelty: the work approaches the self-supervised learning strategy in a novel way favoring pixel-level semantic grouping rather than image-level or object-level representation learning pre-text tasks. The design of the learning procedure has its merit, and the authors state clearly similarity, inspirations from and differences with other methods (e.g. DINO)
- technical soundness: the description of the method is clear and well-argumented, From my perspective, it has a good balance of mathematical formulation, motivations and textual explanation of the several concepts proposed.
- experiments on benchmark datasets: the experimental analysis is done on ImageNet1K and COCO(+), which guarantees that the results can be analysed and commented in full
- the paper is very well-written, clear and concise

*Weaknesses*
- results: results are comparable with those of existing approaches, which have different working principles. Most of the time the slight improvements are marginal (or in some cases the results are also marginally lower than others). This does not allow to fully appreciate the quality and usefulness of the method.
- motivation/background: the main difference with other works is stated to be the fact that no priors related to instance discrimination are used in the pre-text task for self-sup training as they will limit the learning potential. This moves the authors to focus on only exploiting the data intrinsic properties and characteristics. I find this a stretched motivation, as priors related to human knowledge about any problem have been demonstrated to steer the learning process in favor of good performance and efficient use of the data (avoiding to re-learn that prior knowledge somehow from the data).  The fact that this argumentation is weak, is reflected also in the results, which do not show that using only knowledge from the data would contribute to overall better performance (see point above).

---

> ### Author Response · Authors · 2022-08-02
> **Our Response to Reviewer NYjs**
>
> Thanks for your constructive comments. Our responses to them are given below.
>
> ## 1. Clarification on motivation
>
> We would like to clarify that the task of SlotCon is to perform unsupervised object-centric representation learning, with scene-centric datasets as the main objective. Existing methods in this task that rely on hand-crafted priors, are also limited by the prior. For example, ORL that adopt selective-search to find objects, show weaker performance in Table 1; and SoCo that is specilized for detection pretraining, is limited in detection. As the Occam's razor principle goes, entities should not be multiplied beyond necessity. In contrast, SlotCon defines the desirable properties that objects should have, and let the model finds the proper image decomposition by fitting the data. The strength in localizing objects, discovering semantics is supported by experiments (Table 5), and the strength in pretraining with scene-centric data is also clear (Table 1).
>
> ## 2. "Marginal improvement"
>
> We would like to clarify that SlotCon mainly targets scene-centric data, and thus the results on COCO(+) better show SlotCon's improvement. The result on ImageNet is to show that SlotCon is also stronger than or comparable with approaches optimized for object-centric data, but not to propose a new SOTA on it. And the results on unsupervised semantic segmentation are to qualitatively and quantitively show how well the prototypes bind semantics, we yet also do not aim to propose a new SOTA on it.
>
> ## 3. Error analysis with other methods
>
> To better understand why SlotCon improves over previous methods, we extend the COCO results on Table 1 with AP on different objects scales. It shows that SlotCon surpasses the previous SOTA PixPro mainly due to its ability to locate small objects (25.6 vs 24.4 for AP$_\text{s}^\text{b}$).
>
> | Method  | AP$^{\text{b}}_{\text{s}}$ | AP$^{\text{b}}_{\text{m}}$ | AP$^{\text{b}}_{\text{l}}$ | AP$^{\text{m}}_{\text{s}}$ | AP$^{\text{m}}_{\text{m}}$ | AP$^{\text{m}}_{\text{l}}$ |
> | ------- | -------------------------- | -------------------------- | -------------------------- | -------------------------- | -------------------------- | -------------------------- |
> | PixPro  | 24.4                       | 43.5                       | 52.0                       | 18.2                       | 38.9                       | 51.5                       |
> | DetCon  | 23.8                       | 43.1                       | 51.1                       | 17.6                       | 38.3                       | 50.8                       |
> | SlotCon | 25.6                       | 43.8                       | 52.1                       | 18.7                       | 39.2                       | 51.7                       |
>
> ## 4. Limitation discussion
>
> The limitation analysis locates at L154-162 of the supplementary. And the most suitable scenario for SlotCon is self-supervised visual representation learning on scene-centric data. We will try to make it clearer in the revised version.

---

> > ### Comment · Reviewer_NYjs · 2022-08-08
> > **Thanks!**
> >
> > I thank the authors for answering all my questions and doubts. They discussed appropriately almost all my comments.
> >
> > I would, however, further comment on the strong critique of priors, which is supported by the authors with examples in which certain priors are not contributing to an increase of performance or train better models, or provide any particular benefits. Relying completely on data without any prior knowledge can, on the other side, be very dangerous. For instance, in [1] it was shown that relying only on ImageNet data to train a classifier, easily results in models that are bias towards sexist and racist predictions. My point was indeed contrasting the absolute statement that prior knowledge is necessarily "bad" to train models.
> >
> > I am, nevertheless, happy with the response of the authors, and keep my already positive score.
> >
> > [1] J. Zou and L. Schiebinger, “Ai can be sexist and racist — it’s time to make it fair,” Nature, vol. 559, pp. 324–326, 07 2018.

---

> > > ### Author Response · Authors · 2022-08-08
> > > **Thanks for helping improve our paper**
> > >
> > > Dear Reviewer NYjs,
> > >
> > > Thank you for the time you devoted to reviewing this paper and your further constructive comments in the prior issue.
> > >
> > > After carefully reading your comments and reflecting on the experiment results, we agree that our previous statements regarding the priors are too strong and the scope is not well-specified. Here we would like to make the following clarifications:
> > >
> > > * The motivation of SlotCon is more to explore "can we learn good object-centric representations without hand-crafted objectness priors?". This exploration led us to a much simpler architecture that performs yet even better. However, we agree that our critique of prior should be limited to our problem setting (self-supervised pretraining on scene-centric data), and the scope of priors should be limited to objectness priors. Things can vary much if the setting and scope are not specified.
> > > * The use of prior can be viewed as a way of injecting human knowledge into parametric models. Sometimes this kind of knowledge avoids data-guided bias (as in your example), while sometimes, it can limit the model's upper bound or generality (as in our examples). It is impractical to comment generally on whether priors are good or bad. In fact, the geometric-covariance, photometric-invariance, and small cluster number are also helpful priors that we used to train SlotCon and guide it to learn to discover objects/semantics. We need to limit the scope when talking about the influence of priors.
> > > * The data itself can be not reliable. Yes, as pointed out by the reviewer, totally relying on the data may lead to "bad" biases. Although our experiments are limited to scene-centric data, we also notified similar phenomena (detailed in sec. 8 of our response to Reviewer oHQN). The model allocates more prototypes to human-related concepts (as humans occur most frequently in COCO), while many other kinds of animals only have one prototype. When pretraining on a more long-tailed and less discriminative scenario (e.g., autonomous driving data, detailed in our response to Reviewer LB2h), the data may lead the model to learn highly biased prototypes and representations, and harm downstream performance. In this case, human priors adopted for long-tail settings may help.
> > >
> > > Overall, we believe the discussion with the reviewer helps improve our understanding of the problem, clarify the biased statements, and discover possible limitations of the model. We will incorporate the discussions into the next version. Thank you again!
> > >
> > > Best regards,
> > >
> > > Paper 271 Authors

---

### Official Review · Reviewer_oHQN · 2022-07-11

**Rating:** 7
**Confidence:** 3
**Soundness:** 2 fair
**Presentation:** 3 good
**Contribution:** 2 fair

**Summary:**

The paper proposes a method for learning objectness from visual data by combining deep clustering in feature space and a contrastive objective that is made invariant to geometric augmentations.

The method involves two losses:
- $L_\text{group}$ which operates per-pixel and that encourages each pixel to be clustered according to the clusters produced by an EMA-updated teacher,
- $L_\text{slot}$ which operates at the slot level and ensures consistency between slot representations in different augmentations of the same image through a contrastive objective.

After self-supervised pre-training, the encoder is used as the backbone for several vision tasks, namely object detection, semantic segmentation and instance segmentation. In all cases, fine-tuning the pre-trained model yields better results than starting from a random initialization, provided that the number of fine-tuning epochs is equal for the pre-trained and from-scratch models.

**Questions:**

It would be interesting to discuss statistics about the binary indicator in eq. 5 and 6:
- How many slots are active on average for each image? This is mentioned in the supplementary for estimating FLOPS but not discussed from the perspective of representation learning.
- How often is one slot active over the whole dataset?
- How many terms are excluded in eq. 6 because either $1_\text{teacher}$ or $1_\text{student}$ are 0?
- Is there any regularization to ensure that all slots are used evenly? Or to ensure a min/max number of slots per image?

L226: Why the number of prototypes is 256 for COCO and 2048 for ImageNet? Intuitively COCO contains more diverse objects than ImageNet and should use more prototypes.

L281: the text mentions that the method "successfully localizes small objects" w.r.t. column 5 of the figure in table 5. However, that colum depicts a laptop on a bed that are recognized as a small piece of "sky" on the "floor". How come the method has learned to distinguish small portions of pixels but hasn't learned much about semantics? Shouldn't the two losses in eq. 8 optimize for semantic consistency?

Is there anything in the method that explicitly biases the optimization towards learning object-centric representations as opposed to parts?
If not explicitly, why do you think the model learns to group together entire objects rather than parts? Alternatively, do you have evidence that no preference is made between objects and parts?

**Limitations:**

Yes, but only in the supplementary material.
If possible, I would appreciate a mention of limitations in the main text with a link to the corresponding section in the appendix.

**Strengths And Weaknesses:**

The paper is well written and easy to follow, and I am overall positive about it.
The structure of the text introduces the method with gradual complexity, which helps navigating the multitude of symbols and variables.
Although figure 1 appears confusing at first because it has "too much going on", it becomes progressively clearer with the contents of section 3.1 and 3.2.
Most details are not given in the main text, but the supplementary material goes into great detail about datasets and training schedules.

In terms of originality, the paper fits into the popular category of deep clustering methods.
This category has recently seen a shift from image-centric clustering to a finer level of detail.
The strength of this method is to explicitly optimize for desirable properties of clusters, rather than trying to extract clusters from models trained for another objective.

Indeed, the formulation of $L_\text{group}$ resembles a per-pixel extension of DINO because of the prototype-based unsupervised clustering.
Thus, I am happy that the authors anticipated this discussion at the end of Section 3.1.
I agree with the distinction that DINO uses a large number of prototypes while this method uses a smaller number of per-object prototypes.
My intuition is that DINO prototypes capture scene semantics that can be more complex than patch semantics due to compositionality, hence the need for more prototypes.
I am not sure, however, how the prototypes of this method can be "adaptive to each image" (L164).
Eq. 1 clearly shows that the assignments for $L_\text{group}$ are obtained by computing a similarity with learned dataset-wide prototypes, same as DINO. It's only in eq. 4 that slots are extracted by pooling image features using prototype-based attention, these slots can be said to be "adaptive". It would be good to clarify the terms "prototypes", "assignments" and "slots".

Regarding the experiments, I appreciate their exhaustiveness and the choice of datasets.
My main complaint is directed at the lack of confidence intervals for the reported metrics.
Considering how much results can vary due to randomness it is important to report (at least) the mean and std of 3 runs to be sure that improvement is due to the proposed method.
If the number of pages is a constraint at least provide extended versions of tab. 2-6 in the appendix.

Last, a critique for the choice of ablation studies.
Of course, it is interesting to study how the method behaves with a different number of prototypes or different weights for the losses.
However, I do not think the chosen ablations target the most important points of the method.
It would be more interesting to ask: are geometric augmentations (and the subsequent inversion) necessary to learn object-centric representations? Would the model learn something different if only color augmentations were employed on two identical crops of each image? What happens when the binary indicator is not used and all slots contribute to the loss? Would an image-level contrastive objective improve or hinder learning object-level prototypes?

Minor:
- L132 revise "two-layer multilayer perceptron"
- L162 "much less prototypes" -> "fewer"

---

> ### Author Response · Authors · 2022-08-02
> **Our Response to Reviewer oHQN (Part 3/3)**
>
> ## 8. What makes for object-centric representations rather than parts?
>
> Our intuition is that the prototype number and the dataset distribution generate a bottleneck for the granularity of groups. We simply define geometric-covariance and photometric-invariance as the guiding signal, and the model is required to decompose a *large* complex dataset into a *small* number of clusters by optimizing the feature space and the cluster centers. The only solution to this problem is to find the objects/parts that are compositional and thus occupy a reasonable number of prototypes. Concerning granularity, it depends on whether it is helpful to solve the problem given the aforementioned constraints. For example, in our COCO setting with 256 prototypes, the model finds that splitting the animals into cats, dots, elephants, etc., is enough and won't further split them. In contrast, for humans (the most occupying category of COCO), as shown in Figure 7, 8 in the supplementary, the model discovers not only human parts but also human-related activities, indicating that parts are more helpful in this scenario and deserve more prototypes.
>
> ## 9. Mention of limitations in the main text
>
> We thank the reviewer for pointing this out, and will revise it in the next version.

---

> > ### Comment · Reviewer_oHQN · 2022-08-04
> > **Thanks for the thorough answers**
> >
> > The rebuttal addresses all of my points and provides additional insights that I hope will be integrated in the final version. My initial review was already quite positive and I confirm my rating.

---

> > > ### Author Response · Authors · 2022-08-07
> > > **Thanks for helping improve our paper**
> > >
> > > The discussion is thorough, constructive, and helpful for improving the quality of our paper. Thank you again for your time.

---

> ### Author Response · Authors · 2022-08-02
> **Our Response to Reviewer oHQN (Part 2/3)**
>
> ### 4.3 Would an image-level contrastive objective improve or hinder learning object-level prototypes?
>
> We tried to add a MoCo-v3 style instance-level contrastive learning loss to SlotCon (COCO 800 epoch pretraining setting), and the experiment results on COCO show that it depends on the batch size. With a smaller batch size 512, the detection AP drops by 0.2 points, while with a higher batch size 1024, the detection AP raises by 0.4 points. Our explanation is that the instance-level objective is more sensitive to the batch size, and it requires a larger batch size to learn holistic representations that are complementary to the object-level objective. Besides, according to the ablation studies in DenseCL and PixPro, the loss weight of the instance-level loss should also be studied, which is out of our current experiment quota.
>
> | Method         | Batch size | AP$^{\text{b}}$ | AP$^{\text{b}}_{50}$ | AP$^{\text{b}}_{75}$ | AP$^{\text{m}}$ | AP$^{\text{m}}_{50}$ | AP$^{\text{m}}_{75}$ |
> | -------------- | ---------- | --------------- | -------------------- | -------------------- | --------------- | -------------------- | -------------------- |
> | SlotCon        | 512        | 41.0            | 61.1                 | 45.0                 | 37.0            | 58.3                 | 39.8                 |
> | SlotCon w/ ins | 512        | 40.8            | 61.1                 | 44.5                 | 36.8            | 58.1                 | 39.5                 |
> | SlotCon        | 1024       | 40.7            | 61.0                 | 44.4                 | 36.7            | 58.0                 | 39.4                 |
> | SlotCon w/ ins | 1024       | 41.1            | 61.5                 | 45.0                 | 37.0            | 58.7                 | 39.8                 |
>
> ## 5. Statics about the binary indicator
>
> ### 5.1 How many slots are active on average for each image?
>
> It depends on the number of categories/semantics per image. As in Figure 1 of the supplementary file, seven slots are active on average for one image after convergence.
>
> ### 5.2 How often is one slot active over the whole dataset?
>
> It depends on the category/semantic distribution of the dataset, as the slots are roughly bound to real-world semantic categories. We studied the activeness of the slots over the COCO val2017 set that contains 5,000 images, and found that 40 out of the 256 slots are dead, and not active to any image. The activeness of the remaining slots follows a long-tailed distribution. The top-5 active slots correspond to tree (376), sky (337), streetside car (327), modern building exterior wall (313), and indoor wall (307); and the bottom-5 active slots correspond to skateboarder (44), grassland (45), train (56), luggage (57), and airplane (57).
>
> ### 5.3 How many terms are excluded in eq. 6 because either $1_\text{teacher}$ or $1_\text{student}$ are 0?
>
> Given the slot number set as 256, if all slots are active for one image and the two crops overlap well, there should be 256 positive pairs. We studied a well-converged model and found that on average only around 3.6 pairs are active, so around 252.4 terms are excluded. It is reasonable considering that around 7 slots are active on one image, and the overlapping area between the two crops can be small. Besides, considering a total batch size of 512, the number of excluded negative samples should be around 512 x (256 - 7) = 127488.
>
> ### 5.4 Is there any regularization to ensure that all slots are used evenly?
>
> As stated at L152-156, the grouping loss by design avoids two types of collapsing: one slot dominates all pixels, or all slots contribute evenly to every pixel.
>
> ## 6. Why the number of prototypes is 256 for COCO and 2048 for ImageNet?
>
> It is basically an empirical conclusion that setting the number of prototypes close to the number of human-annotated categories can help downstream performance, with detailed discussion at L289-292. It should be noted that though one COCO image is more complex than that from ImageNet, the number of semantic categories that the whole COCO dataset covers is much smaller than ImageNet.
>
> ## 7. Semantic mismatch about unsupervised semantic segmentation
>
> We visualized the prototype that finds the laptop, and found the nearest neighbours are also laptops, which means that the cluster is semantic-cosistent. The problem lies in hungarian matching, where each prototype is assigned to a semantic category. This process adopts a criterion that maximizes the overall pixel accuracy considering the overall performance of all categories. As the semantics of the prototypes are not perfectly aligned with the category labels, it can assign wrong semantic categories to some prototype.

---

> ### Author Response · Authors · 2022-08-02
> **Our Response to Reviewer oHQN (Part 1/3)**
>
> Thanks for your constructive comments. Our responses to them are given below.
>
> ## 1. Comparison with DINO
>
> We are grateful for the reviewer's explanation that scene-level semantics are more complex and thus require more ptototypes, while object-level semantics only need a small number of prototypes due to compositionality. We find it inspiring and will incropate it into the next version.
>
> ## 2. Notation clarity
>
> We will clarify the terms "prototypes", "assignments" and "slots" better in the next version.
>
> ## 3. Error bars
>
> We understand that reporting an error bar for all experiments could make the reported metrics more reliable. However, pretraining is too computation-consuming and thus an error bar is rare to be found from pervious works. Here, we managed to train our model for four independent runs following the settings in Table 1, where our most important results locate. It shows that our method is quite robust across different runs.
>
> | Exp. ID | AP$^{\text{b}}$ | AP$^{\text{b}}_{50}$ | AP$^{\text{b}}_{75}$ | AP$^{\text{m}}$ | AP$^{\text{m}}_{50}$ | AP$^{\text{m}}_{75}$ | City  | VOC   | ADE   |
> | ------- | --------------- | -------------------- | -------------------- | --------------- | -------------------- | -------------------- | ----- | ----- | ----- |
> | No. 1   | 41.03           | 61.13                | 44.97                | 37.03           | 58.32                | 39.80                | 76.24 | 71.62 | 39.00 |
> | No. 2   | 40.92           | 60.99                | 44.62                | 36.79           | 58.13                | 39.42                | 75.84 | 71.72 | 38.94 |
> | No. 3   | 41.09           | 61.20                | 45.04                | 36.89           | 58.26                | 39.56                | 76.17 | 71.49 | 38.34 |
> | No. 4   | 40.97           | 61.03                | 45.08                | 36.98           | 58.02                | 39.84                | 75.86 | 71.72 | 38.66 |
>
> ## 4. Ablation studies
>
> We agree that besides analyzing the hyper-parameters's affect on downstream task performances, it is a good suggestion do dig into the factors that contribute to object-centric representations. We thank the reviewer for pointing this out.
>
> ### 4.1 Are geometric augmentations necessary to learn object-centric representations?
>
> Yes, geometric augmentations are necessary to learn object-centric representations. We made an ablation to train SlotCon on COCO for 800 epochs with two identical crops applied for each image, thus only photometric-invariance is adopted as the supervision. We then visualized the slots the same way as Figure 2, and found that almost none of the slots can bind to a meaningful semantic. Most of them attend to a similar-shaped region that locates at the same position across different images, yet these regions have diverging semantics. And some of them learns textures like animal fur, cloudy sky, snowland, or leaves. A fast evaluation on PASCAL VOC semantic segmentation shows a significant performance drop, yet the representation is still better than random initialization.
>
> | Method                     | mIoU        |
> | -------------------------- | ----------- |
> | Random init.               | 39.5        |
> | SlotCon                    | 71.6        |
> | SlotCon w/o geometric aug. | 62.6 (-9.0) |
>
> ### 4.2 What happens when the binary indicator is not used and all slots contribute to the loss?
>
> The binary indicator is necessary to perform object-level contrastive learning. Omitting it could drastically increase the computational cost and make it infeasible to train the model.
>
> It should be noted that the contrastive learning objective ($\mathcal{L}^\text{Slot}$) is mainly for object-level representation learning based on the slots. Omitting it (and of course also the binary indicator) does not harm the clustering objective ($\mathcal{L}^\text{Group}$), and the model can still learn meaningful slots.

---

### Official Review · Reviewer_MdX5 · 2022-07-11

**Rating:** 5
**Confidence:** 4
**Soundness:** 3 good
**Presentation:** 3 good
**Contribution:** 2 fair

**Summary:**

The paper proposes a self-supervised learning framework, SlotCon, from unlabeled scene-centric data. The method adopts joint semantic grouping -- softly assigning pixels to learnable prototypes shared by the datasets, and contrastive learning -- first perform attentive pooling on the prototypes to from slots, and then conduct contrastive learning on slots from two different views. In experiments, the authors evaluated on COCO, ImageNet-1K and COCO+ dataset. On transfer learning tasks (COCO detection, COCO segmentation, cityscape & Pascal VOC & ADE20k semantic segmentation), the proposed method achieves performance on par or surpassing previous methods. It also achieves better mIoU on unsupervised semantic segmentation results compared with previous methods.

**Questions:**

- How does the proposed method perform on image classification tasks? e.g., ImageNet.


**Limitations:**

The authors addressed the limitations and negative societal impact in the supp.


**Strengths And Weaknesses:**

Strengths:
- The method is purely data-driven without the need of hand-crafted priors or specialized pretext tasks.
- The proposed method perform self-supervised learning on pixel level, which is shown to perform better on dense prediction downstream tasks compared with image-level self-supervised learning methods.
- Extensive experiments are conducted to show the effectiveness of the proposed methods.
- The paper is relatively easy to follow.

Weaknesses:
- Limited novelty and lack of citation on related works. Pixel-level clustering with contrastive learning are already studied in previous works, e.g., [1]. Prototype-based semantic segmentation are also already studied, e.g., in [2]. But many related works are not discussed.

- The representation in the method section is not very clear. e.g., is $A^l_{\theta}$ (Eq. 4) identical to $P^l_{\theta}$ (Eq. 1)? Eq.6 is unclear, since $q_\theta$ is not introduced. Whether it's contrastive learning among two views of a single image, or among a batch is not clear from the equation.
Some symbols in the Method Section are not explained.

- The experiments only show performance on ResNet50, whether the proposed method scales to larger backbones is not clear.

- The COCO detection task is only trained and compared with on 1x schedule, while recent study show that 1x schedule is far from convergence. Hence the results are not very conclusive, it's likely that the proposed method only converges faster, but after training longer, the performance will become similar with comparison methods.

- Type: L211 "solarization".

[1] Ke, Tsung-Wei, Jyh-Jing Hwang, and Stella X. Yu. "Universal weakly supervised segmentation by pixel-to-segment contrastive learning." ICLR 2021.
[2] Zhou, Tianfei, et al. "Rethinking Semantic Segmentation: A Prototype View." CVPR 2022.

---

> ### Author Response · Authors · 2022-08-02
> **Our Response to Reviewer MdX5**
>
> Thanks for your constructive comments. Our responses to them are given below.
>
> ## 1. Lack of related works and novelty concerns
>
> We thank the reviewer for pointing out these related works in pixel-level clustering by contrastive learning. After carefully reading through these papers, we find that there are several key differences between SlotCon and them:
>
> * Setting: SlotCon targets *unsupervised* representation learning (pretraining), while the mentioned works target (weakly-)*supervised* semantic segmentation. While semantic segmentation only cares about the segmentation performance on the source dataset, the performance on various downstream tasks and datasets counts most in pre-training.
> * Motivation: The start point of SlotCon is to learn object-centric representations from unlabeled scene-centric images. Towards this target, it adopts pixel-level clustering for object discovery and builds a contrastive learning objective upon the discovered object-centric representations (slots) to optimize the discriminability of features. Clustering/segmentation is a proxy to learn good features, but not the target.
> * Method: In SlotCon, the clustering process does not require any supervision, and thus the formulation of the learning target is distinct from the mentioned methods. Besides, the clustering is performed with low resolution, and only a coarse objectness estimation is required.
>
> We will include this discussion in the revised version.
>
> ## 2. Symbol clarity
>
> Yes, $\mathcal{A}\_{\theta}^{l}$ in Eq. 4 is identical to $\mathcal{P}\_{\theta}^{l}$ in Eq.1. We adopt $\mathcal{A}\_{\theta}^{l}$ in Eq.1 to cater to the common practice for the cross-entropy loss, and adopt $\mathcal{P}\_{\theta}^{l}$ in Eq.1 to represent assignment/attention.
>
> The $q_{\theta}$ in Eq.6 stands for the predictor, as stated at L188. And the contrastive learning is performed among a batch, as explained at L187-188.
>
> We are sorry for the confusion in symbols and equations caused by inconsistency and unclarity and will revise this part in the next version.
>
> ## 3. COCO detection results with larger backbone
>
> We thank the reviewer for pointing out this issue. In the following table we show the results of SlotCon with ResNet-101 backbone pretrained on COCO for 800 epochs and finetuned with the $1\times$ schedule.
>
> | Method  | Backbone | AP$^{\text{b}}$ | AP$^{\text{b}}_{50}$ | AP$^{\text{b}}_{75}$ | AP$^{\text{m}}$ | AP$^{\text{m}}_{50}$ | AP$^{\text{m}}_{75}$ |
> | ------- | -------- | --------------- | -------------------- | -------------------- | --------------- | -------------------- | -------------------- |
> | SlotCon | R-50     | 41.0            | 61.1                 | 45.0                 | 37.0            | 58.3                 | 39.8                 |
> | SlotCon | R-101    | 42.6            | 62.7                 | 46.7                 | 38.3            | 59.8                 | 41.0                 |
>
> ## 4. COCO detection results with longer schedule
>
> We thank the reviewer for pointing out this issue. In fact the results with $2\times$ schedule is available at Table 1 of the supplementary. Here we extend it for better evaluation. It shows that the performance gain of SlotCon is still significant with a longer finetune schedule.
>
> | Method      | Sche.     | AP$^{\text{b}}$ | AP$^{\text{m}}$ | Sche.     | AP$^{\text{b}}$ | AP$^{\text{m}}$ |
> | ----------- | --------- | --------------- | --------------- | --------- | --------------- | --------------- |
> | **IN-sup.** | $1\times$ | 39.7            | 35.9            | $2\times$ | 41.6            | 37.6            |
> | **PixPro**  | $1\times$ | 40.5            | 36.6            | $2\times$ | 42.2            | 38.1            |
> | **SlotCon** | $1\times$ | 41.0            | 37.0            | $2\times$ | 42.6            | 38.2            |
>
> ## 5. Image classification results
>
> We thank the reviewer for pointing out this issue. In fact SlotCon mainly targets dense prediction tasks like object detection or semantic segmentation. Adding an instance-level loss like DenseCL and PixPro can further contribute to image classification results. Due to the limit in time and computational resources, currently we haven't finished the experiments on ImageNet. Following the setting of [56], with SlotCon pretrained on COCO for 800 epochs with batch size 512, we show the following results:
>
> | Method        | VOC  | CIFAR10 | Cars | Food | Pets | SUN  |
> | ------------- | ---- | ------- | ---- | ---- | ---- | ---- |
> | SlotCon       | 84.2 | 76.6    | 18.9 | 63.0 | 51.2 | 82.3 |
> | SlotCon + ins | 85.9 | 75.9    | 24.6 | 70.0 | 60.4 | 86.4 |

---

> > ### Comment · Reviewer_MdX5 · 2022-08-08
> > **Thanks for the rebuttal**
> >
> > I'd like to thank the authors for answering all my questions.
> > The rebuttal resolves some of my concerns.
> >
> > My main concern for the submission is 1) limited novelty and 2) with longer schedule, the performance is closer to PixPro.
> > I'll keep my previous rating.

---

> > > ### Author Response · Authors · 2022-08-08
> > > **Thank You for Your Time**
> > >
> > > Dear reviewer MdX5,
> > >
> > > Thank you for your time. We appreciate the efforts and comments, which help improve the paper.
> > >
> > > Regarding the novelty issue, in a recently posted general response, we summarized and emphasized the major contributions of this paper in a more general view, and we strongly suggest taking your time for a look and see if it is helpful. Besides, we would also appreciate it if you could point out which specific part in our previous response regarding the novelty issue confuses you, thus we can better resolve your concern accordingly.
> > >
> > > Regarding the closing performance with PixPro in a longer schedule, our response is four-fold:
> > >
> > > * Indeed, the performance in instance segmentation is close (0.1 in AP); however, the performance of SlotCon is still much better in object detection (0.4 in AP). It should be noted that an improvement of 0.4 AP in COCO object detection is significant in this community.
> > > * Besides object detection, SlotCon is also evaluated extensively in semantic segmentation, where SlotCon surpasses PixPro by 0.7 points in the most challenging setting ADE20K. In our response to Reviewer LB2h, SlotCon is also evaluated with challenging autonomous driving data in both pretraining-finetuning and unsupervised semantic segmentation and shows satisfactory results. Sticking to one result in one specific setting may overlook the whole picture.
> > > * It is reasonable that the performance gap between pretraining methods gradually squeezes with a longer downstream training schedule. In fact, in the pretrain-finetune setup, if the downstream data is adequate and the training length is long enough, even random initialization can catch up with imagenet pretraining [1].
> > > * Currently, COCO detection with the 1x schedule is one common config [2-5] to evaluate the quality of pretrained representations. We agree that this setting may be limited, and a more comprehensive benchmark is required, and the extensive evaluation [MdX5, LB2h, NYjs] of this paper in fact, a response to this limitation and a confirmation of SlotCon's superiority.
> > >
> > > [1] He et al., Rethinking ImageNet Pre-training, ICCV 2019.
> > >
> > > [2] Wang et al., Dense Contrastive Learning for Self-Supervised Visual Pre-Training, CVPR 2021.
> > >
> > > [3] Xie et al., Propagate Yourself: Exploring Pixel-Level Consistency for Unsupervised Visual Representation Learning, CVPR 2021.
> > >
> > > [4] Xie et al., Unsupervised Object-Level Representation Learning from Scene Images, NeurIPS 2021.
> > >
> > > [5] Bai et al., Point-Level Region Contrast for Object Detection Pre-Training, CVPR 2022.

---

### Official Review · Reviewer_LB2h · 2022-07-12

**Rating:** 7
**Confidence:** 4
**Soundness:** 3 good
**Presentation:** 4 excellent
**Contribution:** 3 good

**Summary:**

This paper advances a self-supervised representation learning strategy, dubbed SlotCon, that can learn from scene-centric data by reasoning and processing visual information at the pixel level. This is in contrast with the large majority of the self-supervised methods designed for object-centric data (e.g., ImageNet) where image-level reasoning is sufficient.
Recent methods have addressed scene-centric data, however the authors argue that such approaches are limited by the hand-crafted priors (e.g., objects as superpixels) or too specialized pretext tasks as they can limit their generalization.
SlotCon builds upon the SlotAttention method and uses slots as learnable prototypes for pixels that are thus grouped via the assignments to their corresponding slots. SlotCon is composed of a teacher and student network. The pixel grouping supervision resembles both SWaV (aligning the pixel assignments between teacher and student spatial-aligned pixels from different views) and DINO (student + EMA teacher, centering of teacher logits, different temperatures on student and teacher). This loss encourages grouping of pixels into object-like structures via the learned slots.
In order to discriminate slots that carry the same visual or semantic information across views from other non-informative or redundant slots, SlotCon has also a contrastive objective (InfoNCE loss) at the slot level that encourages similarity between different views of the same slot and discourages similarity of slots from different views and slots from other images.

SlotCon is evaluated on a number of pre-training settings (ImageNet, COCO, COCO+) and downstream tasks (object detection, instance segmentation, semantic segmentation, unsupervised segmentation) and datasets (COCO, Cityscapes, Pascal VOC, ADE20k), with nice performance and results.

**Questions:**


Here are a few questions and suggestions to help improve the paper, that could be potentially addressed in the rebuttal


1) Conduct a comparison of the computational cost for training SlotCon compared to related methods, MOCOv2, PixPro, etc.

2) Clarify results and implementations for baseline methods

3) (Optional) How is SlotCon performing on more complex scene-centric datasets, e.g., autonomous driving datasets?


**Limitations:**

Yes

**Strengths And Weaknesses:**

### Post-rebuttal update

The detailed rebuttal addresses all my questions and offers convincing responses and additional insisghts (across the responses to all reviewers).
I don't have any other questions for the authors and I confirm my positive recommendation for this submission.

==========================================


### Recommendation
This paper advance a nice and effective idea for self-supervised learning from scene-centric images. The approach looks sound and the results are encouraging. I'm overall positive about this work and leaning towards recommending it for acceptance.

### Paper strengths

- _Clarity_: I find this paper is mostly well written and argued. The intuition, reasoning and formalism of the method are rather well explained and overview figure 1 is very nice. Then the authors conduct a number of ablation and sensitivity studies (number of prototypes, loss balancing weights, temperature of the teacher) for better understanding. There are also some interesting qualitative results (Figure 2) visualizing pixels grouped by slots and their nearest neighbors. The approach is convincing.

- _Significance_: The paper addresses a challenging problem related to self-supervised representation learning: how to learn from scene-centric datasets that are not as well balanced as ImageNet, more cluttered and usually with fewer images to learn from. The results reached by SlotCon are good.


- _Quality_: The approach seems technically sound. There are several experiments on different settings and different relevant baselines considered. The authors do not provide the code in the supplementary, but provide rich information about the implementation -- it would be better to release the code though. The supplementary is abundant in additional experiments, ablations and implementaiton details.

- _Originality_: This work combines various good practices from the recent literature (SWAV, DINO, Slot Attention, see my comments from the summary), however the output method does still have originality and is effective. The masking of uninformatives slots (eq. 5) is an interesting idea.


- _Misc_: I appreciate that the authors stick to the lower epoch regime (100-200 epochs) which is more reasonable and allows comparison with a larger spectrum of methods


### Paper weaknesses

Mostly minor concerns:

#### Originality:
- as mentioned above, to me SlotCon is a fairly original work aggregating in an effective manner various good practices and approaches from the literature.
- from the text it can be understood that using SlotAttention for unsupervised pixel-level representation learning is a finding of the authors. However other prior methods have used it for weakly-supervised or  self-supervised learning [a], [b]. Those are different approaches and I don't think a discussion is necessary, however the text should be adjust to better delimitate the contributions and prior works
- the approach is similar in spirit with Odin [31] as both aim to learn representations without human object priors. The qualitative examples in both papers are similar. I think it would be good to acknowledge the relatedness of these concurrent works


#### Scope of experiments:
- COCO is indeed a challenging scene-centric dataset and it's great to see pre-training and evaluation done on it, moving beyond ImageNet
- I think this work would benefit from showing results on more complex settings, like autonomous driving data where the amount of objects per scene can be larger (BDD100K, Cityscapes). Some options could be:
    + self-supervised pretraining on an autonomous driving dataset, e.g., BDD100K[56]
    + self-supervised pretraining on smaller autonomous driving datasets, e.g., Cityscapes, and evaluation on unsupervised semantic segmentation on these datasets, see PiCIE, STEGO[c]

#### Computational cost:
- the authors discuss the computational cost in appendix D and give exact number of FLOPS for forward operation compared to a standard ResNet-50 backbone. It appears SlotCon is 12.3% more expensive in terms of FLOPS.
- However this does not tell us much on how fast are forwards and backward passes and how does SlotCon fare against other methods. It's nice that SlotCon can be trained in fewer epochs, but it would be great to be able to compare also the training times
- To get an idea of more straightforward and easy to visualize and understand format I recommend the time and memory cost studies from MoCov2, OBoW[d] and iBOT[e]


#### Baseline results and implementations:
- In Table 3 (transfer results with ImageNet-1k pre-training), it's not clear where to the scores from DetCon (200 epochs) come from. In the original paper there are some result on a few plots and maybe the authors eye-balled them (e.g., COCO instance segmentation), but not all results can be found in the original paper. The authors don't mention another reference for them. Are these scores reproduced by the authors, taken from a different paper (if so, please cite the corresponding source).
- PixPro results in Table 3 seem lower than originally reported, about 0.8-1.0 for COCO. Do the authors know why?

- The common protocol for COCO downstream as proposed in MoCO uses the default `ROI_BOX_HEAD` configuration with 2 FC layers. PixPro seems to have modified the default configuration of 2FC to 4 convolution layers + 1 FC layer that seems to improve scores. Was this setting used for PixPro or SlotCon?


#### Related work:
- this work mentions a whole lot of relevant works from the literature
- however for the unsupervised semantic segmentation section there are several key methods in addition to the now usual PiCIE and IIC and the very recent ones mentioned by the authors (SegDiscover).
- here are a few suggestions for this area of the literature:[f],[g],[h],[c]


**References:**


[a] T. Kipf et al., Conditional Object-Centric Learning from Video, ICLR 2022

[b] Z. Bao et al., Discovering Objects That Can Move, CVPR 2022

[c] M. Hamilton et al., Unsupervised Semantic Segmentation by Distilling Feature Correspondences, ICLR 2022

[d] S. Gidaris et al., Online Bag-of-Visual-Words Generation for Unsupervised Representation Learning, CVPR 2021

[e] J. Zhou et al., iBOT: Image BERT Pre-Training with Online Tokenizer, ICLR 2022

[f] J. Hwang et al., Segsort:Segmentation by discriminative sorting of segments, ICCV 2019

[g] M. Chen et al., Unsupervised object segmentation by redrawing, NeurIPS 2019

[h] X. Wang et al., FreeSOLO: Learning to Segment Objects without Annotations, CVPR 2022

---

> ### Author Response · Authors · 2022-08-02
> **Our Response to Reviewer LB2h (Part 2/2)**
>
> ## 6. Results on autonomous driving data
>
> ### 6.1 Pretraining on BDD100K
>
> As requested by the reviewer, we show the results with BDD100K pretraining and evaluated on Cityscapes semantic segmentation. The model is trained on BDD100K for 800 epochs with 64 prototypes. The result is notably weaker than its COCO counterpart, yet still surpasses MoCo v2 pretrained on COCO. Tough the hyper-parameters might not be well tuned, the BDD100K dataset is indeed challenging for pretraining as its images are less discriminative, and the pretraining on autonomous driving data is a valuable direction to explore. We thank the reviewer for pointing it out.
>
> | Dataset | Method       | mIoU |
> | ------- | ------------ | ---- |
> | -       | Random init. | 65.3 |
> | COCO    | MoCo v2      | 73.8 |
> | COCO    | SlotCon      | 76.2 |
> | BDD100K | SlotCon      | 73.9 |
>
> ### 6.2 Pretraining on Cityscapes
>
> As requested by the reviewer, we show the results pretrained on Cityscapes for 800 epochs with 27 prototypes, and evaluated on unsupervised semantic segmentation. The results are also notably weaker than PiCIE, but surpass MaskContrast and IIC in mIoU. It should be noted that we do not aim to propose a new SOTA for unsupervised semantic segmentation, SlotCon is trained with a much lower resolution from scratch, while the compared works adopts a pretrained model and output high-resolution results. We show these results just to analyze how well the prototypes bind semantics qualitatively and quantitively.
>
> | Method       | mIoU  | pAcc  |
> | ------------ | ----- | ----- |
> | MaskContrast | 3.14  | 40.22 |
> | IIC          | 6.35  | 47.88 |
> | PiCIE        | 10.29 | 72.13 |
> | SlotCon      | 8.92  | 27.95 |
>
> [1] Caron et al., Deep Clustering for Unsupervised Learning of Visual Features, ECCV 2018.

---

> > ### Comment · Reviewer_LB2h · 2022-08-05
> > **About autonomous driving data results**
> >
> > Thank you for these additional results. I find them quite encouraging, given that there was surely no time for tuning.
> >
> > I have just a remark regarding the results from Cityscapes pre-training. PiCIE and IIC (from the PiCIE implementation) use a particular config for pre-training on this dataset: they use do not use only the original train dataset, but also the test set and the remaining coarsely annotated images, summing about to more than 20k images.
> > I suspect the authors pretrained only on the original 3k train images from Cityscapes, which may reduce the performance of SlotCon.
> > Even so, the results are not bad at all!

---

> > > ### Author Response · Authors · 2022-08-07
> > > **Thank your for pointing this out**
> > >
> > > Thank you for pointing out the details of PiCIE's Cityscapes setting. We will try it out in the future.

---

> ### Author Response · Authors · 2022-08-02
> **Our Response to Reviewer LB2h (Part 1/2)**
>
> Thanks for your constructive comments. Our responses to them are given below.
>
> ## 1. Code availability
>
> Our code will be released as promised in the checklist.
>
> ## 2. Discussion with prior works
>
> ### 2.1 Relationship with prior object-centric methods
>
> Learning object-centric representations (object discovery) from unlabelled images has long been the pursuit of the object-centric representation learning community. Directly extracting objects from images is hard due to the lack of supervision and thus prior works have long been restricted to synthetic data.
>
> The recent works [a, b] take the first step to real-world video data, yet they adopt motion flow as a shortcut for objectness prior to solve this problem, and still follow the philosophy of directly extracting objects from an image. In contrary, SlotCon shows that we can first explicitly optimize for desirable properties of clusters (that can describe an object) over the dataset, then retrieve the objects from an image with the learned cluster centers. (This superiority in philosophy is also recognized by Reviewer oHQN.)
>
> More importantly, we first show the possibility of learning object-centric representations from large-scale unlabelled natural scene-centric images in the wild.
>
> Minor correction: SlotCon do share similar spirit with SlotAttention in the competition mechanism between slots, but they are two distinct models in architecture. SlotCon only consists of a set of prototypes to learn, while SlotAttention is a multiple-layered transformer-like model.
>
> ### 2.2 Lacking detailed discussion with concurrent work Odin
>
> We agree that Odin is a highly related concurrent work. Both Odin and SlotCon perform clustering on the feature map to segment objects. However, while Odin just applies kmeans on feature maps to generate masks which are further used to construct the constrastive objective, SlotCon starts from a set of prototypes shared by all samples, which attaches consistent semantic meanings to each cluster at the initial stage and can be adapted to different images to extract image-specific slots.
>
> It is hard to compare with Odin in performance as their reported setting is too computational heavy (trained on ImageNet for 1,000 epochs with batch size 4096 over 128 Cloud TPU v3 workers), but the superiority of deep clustering over kmeans clustering has been shown in the literature [1]. We will make the discussion clearer in the next version of this paper.
>
> ## 3. Lacking discussion with some literature in unsupervised semantic segmentation
>
> We will add the discussion with these papers in the next version.
>
> ## 4. Straightforward computational cost comparison
>
> As requested by the reviewer, here we give a direct computational cost comparison between SlotCon and two previous works. The experiments are conducted on the same machine with 8 NVIDIA GeForce RTX 3090 GPUs. Both PixPro and SlotCon adopt a batch size of 1024 and have amp turned on, and DenseCL adopts a batch size of 256 by default. The training time of DenseCL might be higher as we failed to install apex.
>
> | Method  | Time/epoch | Memory/GPU |
> | ------- | ---------- | ---------- |
> | DenseCL | 2′46′′     | 7.9GB      |
> | PixPro  | 2′19′′     | 15.1GB     |
> | SlotCon | 2′23′′     | 16.0GB     |
>
> ## 5. Details about baseline results and implementations
>
> ### 5.1 Source of DetCon results in Table 3
>
> Yes, as anticipated by the reviewer, these results are eye-balled from Figure 4 in the DetCon paper. The COCO detection result is taken from their 1st version on arXiv (https://arxiv.org/pdf/2103.10957v1.pdf), which was deleted in their 2nd version (https://arxiv.org/pdf/2103.10957v2.pdf). And other results are taken from Figure 4 in the `v2` paper. We apologize that some results should be corrected in Table 3 concerning DetCon: the COCO detection result should be 40.6 rather than 40.5, and the Cityscapes result should be 75.5 rather than 76.5. We'll make the data source clearer and correct the results in the next version.
>
> ### 5.2 Lower PixPro results in Table 3
>
> The highest IN-100ep result reported in Table 1 of PixPro (AP 41.3) is produced by pretraining with a FPN (see their Section 3.3 and Table 2(e) for details). This setting is, however, not adopted in their released code & models (https://github.com/zdaxie/PixPro), on which our re-implementation is based. We'll clarify this in the next version.
>
> ### 5.3 COCO downstream protocol
>
> Our COCO downstream implementation is directly copied from PixPro, so yes we also adopt the 4 convolution layers + 1 FC layer configuration. This setting can trace back to InfoMin (https://github.com/HobbitLong/PyContrast/tree/master/pycontrast/detection), which is also widely adopted (e.g., InsLoc, DenseCL, PixPro, ORL, SoCo according to their official codebase). As all results reproduced by us adopt the same COCO downstream config with SlotCon, this won't harm fair comparison. We'll clarify this in the next version.

---

> > ### Comment · Reviewer_LB2h · 2022-08-05
> > **Post-rebuttal comment**
> >
> > I would like to thank the authors for the (very) detailed rebuttal.
> > The rebuttal addresses all my questions and offers convincing responses and additional insisghts (across the responses to all reviewers).
> >
> > I don't have any other questions for the authors and I confirm my positive recommendation for this submission.
> > Nice work!
> >
> > Just a comment regarding a potential misunderstanding of some of my comments:
> > - response 2.1: On the use of SlotAttention. As mentioned before, I find that [a,b] are different approaches from SlotCon. My initial comment was regarding the phrasing of the use of the SlotAttention principle (I'm aware about the architecture and implementation differences). There are no concerns from side on the quality of this work; the suggestion was to better emphasize the contribution of this work. I think that a compressed variant of the author's answer would be useful in the paper.
> > - response 2.2: Thanks for the detailed answer about Odin. I did not ask for a comparison with Odin as they are concurrent works. I've just mentioned that they are aiming for similar things, though by different means. Again, there is no concern from my side regarding Odin and SlotCon

---

> > > ### Author Response · Authors · 2022-08-07
> > > **Thanks for helping improve our paper**
> > >
> > > The discussion is definitely constructive and produces new insights. We would like to thank the reviewer for helping improve our paper.

---

### Author Response · Authors · 2022-08-02
**General Response**

We thank all the reviewers for their unanimously positive reviews and insightful comments.

We are glad to find that our paper is consistently considered to be well written, clear, and concise [LB2h, MdX5, oHQN, NYjs].

We also appreciate that the reviewers think our method is nice and effective [LB2h, MdX5], and is a convincing approach [LB2h], and the uses of our method are very broad [NYjs].

Individual concerns have been addressed carefully in the response to each reviewer.

We will revise the paper following the suggestions.

---

### Author Response · Authors · 2022-08-08
**Post-rebuttal Response from the Authors**

Dear Reviewers,

The discussion with each reviewer is greatly constructive and helps understand SlotCon from a specified perspective. To better show the full picture, we would like to re-emphasize the significance of this paper in a more general view.

SlotCon mainly sheds light on two important subfields of representation learning: self-supervised representation learning (pre-training) and object-centric representation learning (object discovery). In one attempt, it jointly solves two challenging problems in them, as detailed following:

For the pre-training community, how to move beyond the object-centric dataset ImageNet, and *learn from scene-centric datasets* that are *imbalanced*, more *cluttered* and usually *with fewer images to learn from* has been a challenging problem [LB2h]. It is commonly believed that adopting the complex structure (e.g., multiple objects) in scene images can benefit feature learning [1], and how to expose the structure of the data (produce objectness) has been the key to solving this problem. Unlike current approaches that are limited by the hand-crafted priors (e.g., objects as superpixels) or too specialized pretext tasks, *SlotCon shows that directly optimizing a clustering objective is enough to produce satisfactory and even better objectness*, which paves the ground for a contrastive learning objective (InfoNCE loss) at the slot level and help learn significantly stronger representations (evaluated extensively on multiple downstream tasks). During the rebuttal period, we find it also successes in the challenging autonomous driving data, highlighting the generality of SlotCon.

For the object discovery community, it has long been the common pursuit to *learn object-centric representations from unlabelled images*. Directly extracting objects from images is challenging due to the lack of supervision, and thus prior works have long been restricted to synthetic data. Some recent works take the first step to real-world video data, yet they adopt motion flow as a shortcut for objectness prior to solving this problem and still follow the philosophy of directly extracting objects from an image. On the contrary, SlotCon shows that we can first explicitly optimize for desirable properties of clusters (that can describe an object) over the dataset, then retrieve the objects from an image with the learned cluster centers [oHQN]. More importantly, *SlotCon first shows the possibility of learning object-centric representations from large-scale unlabelled natural scene-centric images in the wild*.

SlotCon also shows good ability in unsupervised semantic segmentation. Still, we want to clarify that this is just to help understand the learned prototypes, but not to propose a new SOTA.

In terms of originality, SlotCon does share the spirit with some prior works (e.g., DINO and Slot Attention). In the paper and rebuttal, we have well discussed its relationship with them, and we appreciate that SlotCon is recognized as a fairly original work [LB2h], the design of the learning procedure has its merit [NYjs], and the authors state similarity, inspirations from and differences with other methods clearly [oHQN, NYjs].

[1] Hénaff et al. Object discovery and representation networks, arXiv preprint.

Best regards,

Paper 271 Authors

---

### Meta-Review · Area_Chair_ipSN · 2022-08-29

**Recommendation:** Accept
**Confidence:** Certain

**Metareview:**

This paper proposes an object-centric representation learning based on a data-driven semantic slots from scene-centric data. In specific, the proposed SlotCon simultnesobly performs semantic grouping and contrastive representation learning over groups (slots), which naturally leads to obtaining object-level representations without any prior knowledge. The proposed algorithm is technically sound and novel. It is clearly distinct from the previous dense contrastive learning in that it jointly learns the target grouping of pixels. Most of all, it first shows encouraging performances by object-centric representation learning on natural image datasets. Even though the performance improvements seem to be somewhat marginal in comparison to the previous SOTA algorithms, the proposed method fairly demonstrates the effectiveness and feasibility in the use of object-centric representation learning for scene-centric data and the corresponding downstream tasks. In addition, the authors properly addressed almost all concerns and questions raised by the reviewers. In conclusion, I would like to recommend to accept this paper.

**Award:**

No

---

### Decision · Program_Chairs · 2022-09-14

Accept